# Relationship between rheology and structure of interpenetrating, deforming and compressing microgels

Gaurasundar M. Conley[1], Chi Zhang[1], Philippe Aebischer[1], James L. Harden[2] & Frank Scheffold [1]

Thermosensitive microgels are widely studied hybrid systems combining properties of polymers and colloidal particles in a unique way. Due to their complex morphology, their interactions and packing, and consequentially the viscoelasticity of suspensions made from microgels, are still not fully understood, in particular under dense packing conditions. Here we study the frequency-dependent linear viscoelastic properties of dense suspensions of micron sized soft particles in conjunction with an analysis of the local particle structure and morphology based on superresolution microscopy. By identifying the dominating mechanisms that control the elastic and dissipative response, we can explain the rheology of these widely studied soft particle assemblies from the onset of elasticity deep into the overpacked regime. Interestingly, our results suggest that the friction between the microgels is reduced due to lubrification mediated by the polymer brush-like corona before the onset of interpenetration.

[1] Department of Physics, University of Fribourg, Chemin du Musée 3, 1700 Fribourg, Switzerland. [2] Department of Physics, University of Ottawa, Ottawa, Ontario K1N 6N5, Canada. Correspondence and requests for materials should be addressed to F.S. (email: Frank.Scheffold@unifr.ch)

Soft polymer microgels are fascinating systems whose peculiar properties have resulted in highly diversified applications, spanning from purely academic to the industrial domain[1,2]. Microgels as model soft spheres have been instrumental in shedding light on fundamental problems relating to phase transitions[3–6] and microgel additives as rheological modifiers are ubiquitous in consumer and personal care products as well as other industries and applications[7–11]. The complex nanoscale architecture and softness sets them apart from more conventional solid particles, emulsion droplets or foam bubbles, with profound consequences for the mechanical properties of dense microgel suspensions, which reveal rich and complex features in their concentration dependence[8,12,13].

The rheology of hard spherical particles in suspensions is controlled by the volume fraction $\zeta$ of the dispersed phase as the sole parameter determining a suspension's phase behavior. In a disordered suspension of hard spheres the maximal volume fraction is reached at random close packing or jamming, $\zeta_{rcp} \simeq \zeta_J \simeq 0.64$[4,14,15]. Emulsions, bubbles, and other soft building blocks, on the other hand, can deform, allowing for $\zeta_J \leq \zeta < 1$. In this range the particles form flat facets at contact points which in turn store elastic energy[16–18], resulting in familiar soft pastes such as mayonnaise or shaving foam. Polymer microgels however are different. They are highly swollen in good solvent conditions, and as a consequence, microgels are compressible in addition to being deformable and therefore highly overpacked states can be reached[1,19–21]. Moreover, microgels prepared following standard protocols have a fuzzy polymer shell decorating their compressible cores[22], allowing for shell compression and interpenetration[23]. Much work has been devoted to the characterization of the elasticity and flow of dense microgel suspensions (or pastes) and common features for microgels of different sizes and softness have been established[20,24–31]. The elastic modulus grows rapidly after the liquid–solid transition and then much more slowly at higher concentrations. Relatively little is known, however, about dissipative losses in dense microgel suspensions under shear and their relationship to the microstructure.

Although models and detailed numerical studies have shed much light on microgel rheological properties[26,28,30,32–35], there still exists no widely accepted framework that encompasses the entire range of packing densities, from the glassy dynamics and jamming to highly compressed states. Depending on the type (more or less ionic) and size of microgel, osmotic deswelling[36], and interpenetration can be important, but if and when this plays a role has been debated[23,35,37]. In the past, little or no in situ information on the single-particle nanoscale level has been available. Recently, however, significant progress has been reported in studies revealing single-particle properties in dense suspensions based on zero average contrast small angle neutron scattering[23] and microscopy[34,37–40].

In this work, we propose a framework to explain the frequency-dependent linear viscoelasticity of microgel suspensions, composed of micron-sized poly(N-isopropylacrylamide) (pNIPAM) microgels with a dense core and a fuzzy corona, from weakly compressed packings to strongly overpacked states by combining the results from oscillatory shear measurements and nanoscale imaging. To this end we characterize the macroscopic rheological properties of soft particle suspensions at a constant temperature and take advantage of the microscopic structural information about individual microgels and pairs of microgel particles resolved via dSTORM superresolution microscopy[41]. Our aim is to describe and connect the mechanisms that determine the viscoelastic and in particular the dissipative behavior across the different concentration regimes. From the latter we can derive important information about the lubrified facetted microgel interfaces and the onset of corona interdigitation. We note that we do not address the properties of ionic[42] or weakly cross-linked microgels particles[43] nor of those with radii of less than 100 nm. As discussed in recent work, small particles appear to behave qualitatively differently and can for example be overpacked up to a factor of ten without a corresponding increase in elastic modulus[31,44,45].

## Results

**Superresolution microscopy**. We study swollen pNIPAM microgels prepared by free radical precipitation polymerization at a constant temperature $T = 22\ ^\circ C$. From static and dynamic light scattering we find the total microgel radius is $R_{tot} \simeq 470$ nm and the radius of the highly cross-linked core is $R = 380$ nm[38]. Details about the synthesis and characterization are included in the methods section (additional data is plotted in the Supplementary Figs. 1 and 2). The same batch of microgels was used in our earlier work, ref. [37]. The experimentally accessible mass concentrations $c$ of our suspensions, in wt/wt%, can be converted into effective packing fractions $\zeta = k \times c$ via the voluminosity $k = 0.08$, as shown in ref. [37]. We estimate the error bar in setting $\zeta$ to about $\pm 3\%$, see also ref. [46]. We have verified, using small angle light scattering that, on the time scale of the experiment, the samples do not crystallize[37] (see also Supplementary Fig. 3). The structure and morphology of standard, micron-sized pNIPAM microgels and microgel pairs are resolved via single and dual color super-resolution microscopy and small angle light scattering, from marginally jammed to deeply overpacked states, as depicted in Figs. 1 and 2 and discussed in detail in our earlier work[37,38], see also Supplementary Fig. 3. We have determined the lateral spatial dSTORM resolution to be approximately 30 nm, about an order of magnitude better than conventional widefield light microscopy as shown in Fig. 1. To obtain a faithful contour of each particle and use it for measuring different geometric features we use the Laplacian of Gaussian edge detector in Matlab (MathWorks, Inc., USA) (for details see methods section). Using synthetic data we can estimate the statistical error of the contour line determination to about $\pm 5$ nm or less. We can identify three consecutive stages of packing[37]. In the first stage, just above solidification ($\zeta \gtrsim 0.64$ or $c \gtrsim 8\%$), the microgel's fuzzy corona or brush is compressed as the measured distance between neighboring particles drops below $d = 2R_{tot}$, see Fig. 2 and Supplementary Figs. 3 and 4. We note that the compression of the fuzzy corona cannot be visualized directly with dSTORM, due to the extremely low polymer density and the associated noisy signal in this region (see also Supplementary Fig. 2). In the following stage, once the dense cores come into contact ($\zeta \gtrsim 1.1$), interpenetration becomes noticeable and the microgels start to significantly deform, which allows denser packing of the particles without change in measured size (see also Supplementary Fig. 5). Interpenetration gradually increases as the contacting facets expand. Finally, once interpenetration and deformations have saturated and the volume is homogeneously filled by the polymer gel ($\zeta \gtrsim 1.9$), isotropic compression and a reduction of the microgel size $\propto \zeta^{-1/3}$, remains the only mechanism that allows further densification.

**Oscillatory shear experiments**. We perform oscillatory shear measurements in the linear regime (strain $\gamma = 0.1\%$) at a fixed temperature of $T = 22\ ^\circ C$ covering a wide range of $\zeta$, from the onset of jamming $\zeta \gtrsim \zeta_J = 0.64$ to deeply overpacked, and determine the elastic and loss moduli as a function of frequency, $G'(\omega)$ and $G''(\omega)$. Selected examples of frequency-dependent measurements of $G'$ and $G''$ are shown in Fig. 3a covering the $\zeta$ range from marginally jammed to deeply overpacked. In all cases we find $G'$ being nearly frequency independent and greater than $G''$, indicating solid like behavior. Dissipative losses, however, are

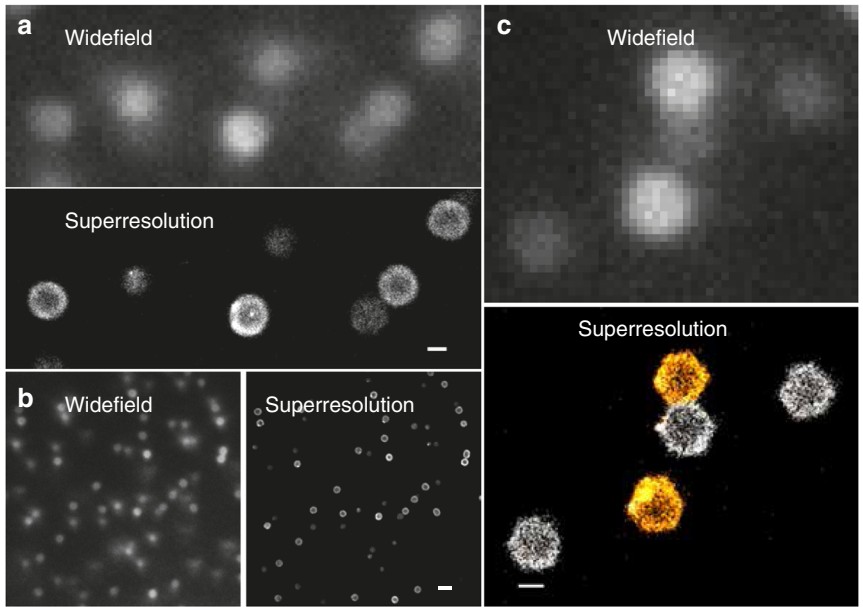

**Fig. 1** Comparison of conventional widefield and dSTORM superresolution images of densely packed microgels. **a**, **b** show images of Alexa Fluor®647 dye-labeled microgels seeded in a dense suspension ($\zeta = 0.86$, $c = 10.8$ wt%). Images are taken in the bulk and thus some of the particles are out of focus and located deeper inside the sample, appearing less bright. This also affects the superresolution imaging leading to a reduced number of localized points that can be used for the reconstruction. **c** Widefield and two-color dSTORM images of microgels at the glass-sample interface including a pair in contact ($\zeta = 1.89$, $c = 23.6$ wt%). The widefield image was taken with a low pass filter for Alexa Fluor®647 and thus the microgel labeled with second dye CF680R is barely visible. The two-color image was reconstructed using the spectral demixing technique[37, 63]. Scale bar 500 nm in panel a and c and 2 μm in panel **b**

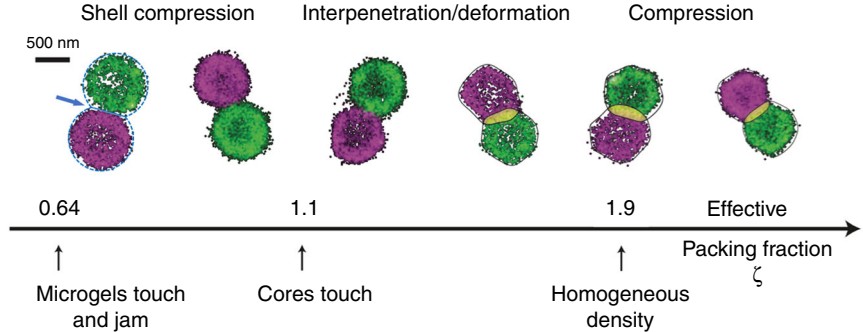

**Fig. 2** Two-color dSTORM superresolution microscopy of microgel pairs. Examples for dye-labeled tracer particles seeded in densely packed microgel suspensions are shown ($T = 22\,°C$)[37]. Particles are labeled with the fluorophores Alexa Fluor®647 or CF680R. The effective packing fraction $\zeta$ increases from left to right ($\zeta = 0.86, 1.01, 1.26, 1.50, 1.89, 2.13$). Left: The dashed circles with radius $R_{tot} = 470$ nm visualize the total microgel size including the barely visible low-density corona. The arrow points to the contact area where the brush-like corona is partially compressed. The straight line indicates the cross-section of the contact area. Right: The solid lines show the contour of the microgels for higher packing densities where the corona has already been fully compressed onto the core and microgels interpenetrate[37]. The overlap area $\Delta F$ is highlighted in yellow

relatively high and $G''(\omega)$ shows a minimum around $\omega \sim 1$ rad/s, typical of emulsions and foams, in addition to microgels[16,29,47]. With increasing concentration, the minimum becomes progressively less pronounced and finally, with $\zeta = 1.9$, has all but disappeared. To characterize the $\zeta$—dependent elasticity of our microgel suspensions we take the value $G'(\omega)$ at a fixed frequency of $\omega = 1.2$ rad/s, Fig. 3b. Below the jamming packing fraction $\zeta_J$ we measure a weak elastic modulus that we can tentatively ascribe to the entropic glass regime, where the onset of elasticity is given by $G_p \sim k_B T/R^3$, the only energy density scale for noninteracting spheres[48] (in our case $k_B T/R^3 \simeq 0.04$ Pa), which then crosses over to a regime governed by the jamming elasticity (see also refs. [33,49] and Supplementary Fig. 6). Starting from $\zeta \sim \zeta_J$, when microgel coronas are in direct contact, we find a steep increase of $G'$. Increasing $\zeta$ by a factor two results in a nearly three order of

magnitude increase of $G'$. This trend, however, does not continue over the entire range, instead we observe a slow crossover into a different regime where the slope is reduced considerably[17,25,28,29,50].

**Elasticity and storage modulus $G'(\omega)$.** In earlier work Senff and Richtering studied neutral pNIPAM microgels chemically similar to ours but of smaller size $R \sim 130$ nm at 20 °C[20]. In this pioneering study the rapid increase of elasticity after jamming has been described ad-hoc in terms of a soft interaction potential of the form $\psi \sim r^{-n}$ [20,25] resulting in a power law $G' \sim \zeta^m$ with $m = 1 + n/3$. A more physically descriptive, yet still quantitative approach has been proposed by Scheffold et al.[28] for micron sized microgels, whereby the microgels are modeled as solid cores, of

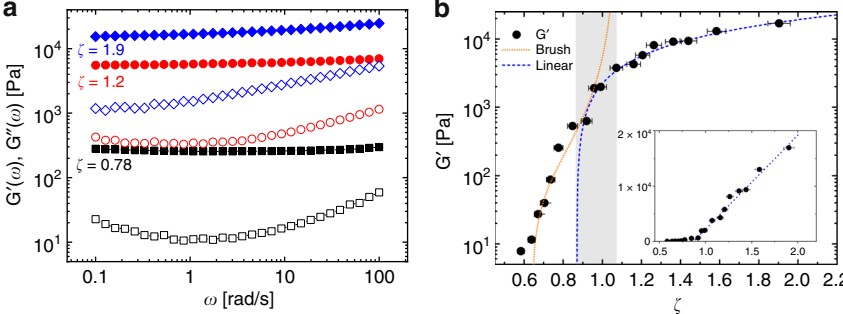

**Fig. 3** Oscillatory shear experiments on dense microgel suspensions. **a** $G'(\omega)$ (full symbols) and $G''(\omega)$ (open symbols) as a function of frequency $\omega$ for different packing fractions, ranging from marginally jammed to deeply overpacked. **b** $G'(\omega = 1.2 \text{ rad/s})$ as a function of packing fraction $\zeta$ fitted with the Brush model (orange dotted line) at lower $\zeta$ and with a linear function $G' = 17 \text{ kPa} \times (\zeta - \zeta_c)$, dashed line for $\zeta \geq \zeta_c = 0.87$. The shaded area marks the crossover between the regime controlled by corona compression to the one governed by core–core interactions. Inset: Linear representation of the same data. The estimated error in $\zeta$ shown in panel (**b**) is ±3%

size $R$, decorated by polymer brushes, of thickness $L_0 \simeq R_{\text{tot}} - R$, which mediate their interactions. The net repulsion between brushes at the microgel periphery, derived from the Alexander—de Gennes scaling model for polymer brushes in good solvent conditions, suffices to describe the onset of solid like behavior. To derive an expression for $G'$, a local spring constant $\kappa$ is defined which is directly related to the interaction potential between two spheres by $\kappa = \partial^2\psi/\partial r^2$, and to the elastic modulus as $G' \approx \kappa/\pi R$. The complete expression for $G'(\zeta,\alpha) \simeq kT\alpha s^3(\tau^{9/4} - \tau^{-3/4})$, where $\tau(\zeta, \alpha) = (1 - \alpha)/[(\zeta/\zeta_J)^{-1/3} - \alpha]$, modulus a prefactor of order unity, is derived in[28] where $s$ is the effective average separation between grafting sites and $\alpha$ is the ratio between core and total radius of the particle. By setting $\alpha = 0.84$ and adjusting the prefactor $k_B T\alpha/s^3 = 60$ Pa for a best fit we obtain the dotted orange line shown in Fig. 3b. The value of $\alpha = 0.84$ compares well with the static light scattering result $R/R_{\text{tot}} \simeq 0.81$ (see Supplementary Fig. 1) and previous studies on similar systems[28]. We find very good agreement with the experimental data in the lower concentration range, $0.64 \leq \zeta \lesssim 1$, delineating the range where the microgels are predominantly interacting via their brush-like coronas, Fig. 2. Instead of the divergence of $G'(\zeta)$, predicted by the brush model we find, at higher concentrations, a slower, linear increase of elasticity as a function of packing fraction. This is in agreement with several previous studies on dense microgel packings and can be attributed to the finite softness of the microgel core[29,31,32,50–52]. By extrapolation, we can estimate an onset of the linear regime at $\zeta_c = 0.87 \pm 0.01$. This occurs well before the divergence predicted by the brush model at $\zeta \approx 1.08$, resulting in a crossover region where the softness of the core eventually dominates over the stiffness of the highly compressed corona, see also the recent work by Bergman et al.[53]. As shown in Fig. 4b, for $\zeta \in [\zeta_c, 1]$ the brush model and the linear scaling superimpose while for $\zeta \geq 1$ the core softness dominates. The fuzzy shell is now compressed almost entirely onto the core. Thus, the linear increase of elasticity is consistent with the jamming of discrete homogeneous microgel particles of size smaller than the unperturbed radius $R_{\text{tot}}$ (see also Supplementary Fig. 4). We note that for dense emulsions, consisting of homogeneous, incompressible soft particles, a linear increase $G' \sim \zeta - \zeta_c$ is also observed[16]. As $\zeta$ is increased further for the microgels, significant particle deformations can be seen by dSTORM, Fig. 2[37]. The entire system now becomes more and more homogeneously filled with polymer as the interstitial spaces between spherical particles in contact vanish. The latter is confirmed by small angle light scattering data[37] (see also Supplementary Fig. 3), which shows a dramatic drop in the scattering contrast for $\zeta \geq 1.9$ ($c \geq 23$ wt%). Interestingly, for these very large filling fractions, we find an

elastic modulus $G' \approx 10^4$ Pa, comparable to data reported by Calvet et al.[54] for macroscopic homogeneous pNIPAM gels of similar composition to ours (~5 mol% BIS).

**Friction and loss modulus $G''(\omega)$.** Next, we consider the energy losses in the system where the influence of the discrete particulate nature of the suspension at high-packing fractions is striking. Over the entire range we find significant dissipative losses, typical for soft glassy materials[55], but in stark contrast to macroscopic gels which are almost entirely elastic in their stress response. Calvet et al.[54] found for macroscopic pNIPAM gels $G''/G' \sim 10^{-3}$, typically about two orders of magnitude less than what we observe.

Anomalously large losses are well known for jammed emulsions, despite the fact there are no static friction forces between the emulsion droplets. Liu et al. showed, that the high-dissipative losses observed for emulsions are due to dynamic dissipation in the fluid confined between planes of facets sliding relative to each other[56]. The random orientation of slip planes leads to a broad range of stress relaxation rates that result in $G'' \sim A(\zeta)\omega^{0.5} + \eta_\infty\omega$ in this regime, where $\eta_\infty$ denotes the background viscosity of the solvent phase. Their amplitudes $A(\zeta)$ increase by about a factor 3–4 over the concentration range accessible for emulsions, $\zeta \simeq 0.6 - 0.86$, which can be explained by the increased viscosity of the compressed liquid film confined in the shear planes[56]. Earlier pioneering work by Cloitre and co-workers on polyelectrolyte microgels, with a radius $R = 220$ and $R = 125$ nm, already suggested that densely packed microgels develop flat facets at contact and that the thin water film trapped in between can lubricate the contacts, but a connection to the model of Liu et al. was not made[17].

Based on the emulsion work and by comparison with the dSTORM data we can now verify the accuracy and the range of validity of this scenario for our microgel system. As discussed before, we described the microgel by a cross-linked core covered by a brush-like corona. At $\zeta > \zeta_J$ the brushes are partially compressed and the restoring forces lead to the rapidly increasing macroscopic shear modulus $G'$, Fig. 3b. It is known that compressed polymer brushes of thickness $L < L_0$ do not interpenetrate and do not show any noticeable friction when sheared slowly against each other, down to compression ratios of $L/L_0 \sim 0.1 - 0.15$[57–59], suggesting that the model developed for emulsions should also apply to microgels, at least over a limited concentration range, before the onset of interpenetration. As a critical test, we fit $G''(\omega) = A(\zeta)\omega^p$ with adjustable parameters $A$ and $p$ roughly over a decade in frequency $\omega \in [10, 100]$ rad/s. Figure 4a shows these fits in a logarithmic representation for each

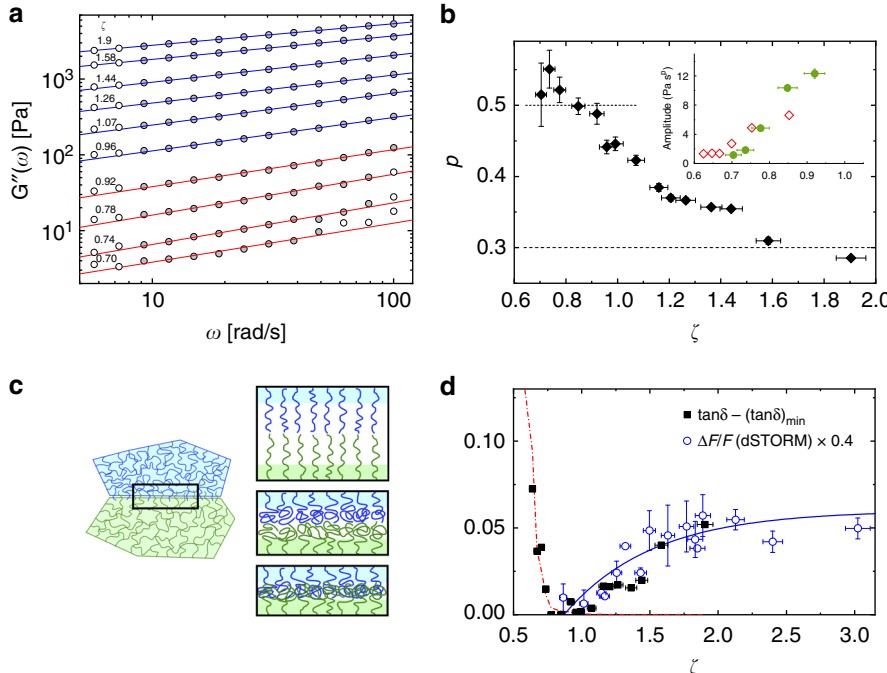

**Fig. 4** Loss modulus, interfacial lubrifiaction and friction in densely packed microgel suspensions. **a** Loss modulus $G''(\omega)$ for frequencies $\omega \geq 4$ rad/s for different packing fractions (symbols). Solid lines show the fit to the data with $G''(\omega) = A(\zeta)\omega^p$. Data points included in the fit range are filled dark gray. Lines with $p \simeq 0.5$ are red, $p < 0.5$ blue. **b** Values of $p$ obtained by a best fit to the data. Inset: concentration dependence of $A(\zeta)$, $\zeta \leq 0.92$. Full symbols show the data for microgels, corresponding to the red lines in **a**, open symbols for emulsions (from ref. [56] with $p \equiv 0.5$). **c** Illustration of corona–corona interactions at the interface between microgels, see also Fig. 2. Initially the dangling corona polymer chains are extended in a brush-like configuration. At higher densities ($\zeta \sim \zeta_c \sim 1$) the chains are compressed. For $\zeta > \zeta_c$ microgels interpenetrate. **d** Dissipative losses $\tan\delta = G''/G'$ at $\omega = 1.2$ rad/s. Solid squares: $\tan\delta - (\tan\delta)_{min}$ with $(\tan\delta)_{min} = 0.043$. Open circles: rescaled overlap area $0.4 \times \Delta F/F$ of adjacent microgels derived from dSTORM (see also Supplementary Fig. 5)[37]. Dash-dotted line denotes $(1/G')$ [arb.u.] and the solid line $\lambda(1 - \exp[-(\zeta - \zeta_c)/\xi])$ with $\lambda = 0.06$ and $\xi = 0.65$ is a guide to the eye. The estimated error in setting $\zeta$ shown in panels **b** and **d** is $\pm 3\%$

choice of $\zeta$. We note that the contribution of the background fluid is small enough that it can be safely neglected over the range of frequencies $\omega \leq 100$ rad/s considered. Up to $\zeta \simeq 0.9$ the data are well-described by the $\omega^{0.5}$ scaling predicted for emulsion droplets with no static friction and also the amplitudes $A(\zeta)$ are similar to those reported for emulsions in ref. [56], inset Fig. 4b. Quantitative differences in $A(\zeta)$ between emulsions and microgels can be explained by the fact that disjoining pressure between the droplet interfaces and the compressed brushes of the microgel corona are not exactly the same.

Figure 4b shows the dependence of the power-law fit parameters $A$ and $p$ on $\zeta$. Starting at $\zeta \gtrsim \zeta_c$ deviations from the $\omega^{0.5}$ scaling can be clearly observed. For a solid core the corona would be entirely compressed on the core at $\zeta \to 1.08$, but in our case the core and the corona deformation are coupled and the transition is smeared in the range $\zeta \in [0.87, 1.08]$ due to the compressibility of the core. Thus for $\zeta \geq \zeta_c$ chains in the corona are not stretched anymore and the density of the corona and the core gradually approach each other[32], as illustrated in Fig. 4c. As a consequence the penalty for the dangling ends of the corona to interdigitate becomes progressively smaller and losses, expressed by the ratio $G''/G' = \tan\delta$, increase over the entire frequency spectrum probed, full symbols in Fig. 4d (see also Supplementary Fig. 6b). In this regime, two-color dSTORM provides key information about this interdigitation process. The open circles in Fig. 4d show the overlap area $\Delta F/F$ extracted from a ~500 nm thick $z$-section through the center of adjacent microgel particles[37] (see also Supplementary Fig. 5). The overlap increases rapidly from $\zeta_c = 0.87$ to $\zeta = 1.9$ and saturates above. We also observe a lowering of the high frequency slope, from $G'' \sim \omega^{0.5}$ to $\sim\omega^{0.3}$,

dashed lines in Fig. 4b, marking a deviation from the viscous behavior of jammed emulsions where the slope of 0.5 is maintained[56], but in agreement with previous observations on dense microgel suspensions[17,27,31]. Interestingly, we find that the anomalously large losses, expressed in terms of $\tan\delta = G''/G'$, scale directly with the overlap area $\Delta F/F$ derived from super-resolution microscopy, as shown in Fig. 4d. In particular, both seem to rise together toward a plateau value at large $\zeta$. We stress the fact that for $\zeta < \zeta_c$ the situation is entirely different. As long as the corona is not yet fully compressed, the brush–brush interfaces of the touching microgel coronas are lubricated, $G''$ increases slowly and thus the relative viscous losses drop with the modulus: $\tan\delta \propto 1/G'$ as shown by the red dash-dotted line in Fig. 4d.

## Discussion

Our investigations reveal that the onset of elasticity in the dense microgel suspensions is governed by compression of their fuzzy outer shells while the friction between the microgels is reduced due to lubrification mediated by this polymer brush-like corona. At higher packing fractions, we visually observe deformation, interpenetration and compression and here the elasticity increases linearly with concentration starting at $\zeta_c$, in agreement with the jamming picture of dense assemblies of homogeneous soft spheres. Deep in the jamming regime our microgel particles have lost their core–shell structure as the soft brush-like corona has already been compressed onto the core. Eventually the faceted and interpenetrating polymeric particles fill space homo-geneously. Interestingly, in contrast to other recent studies, we find no evidence for spontaneous deswelling due to an ionic

osmotic pressure difference between the inside and the outside of the microgels as suggested by Scotti et al.[36]. Revealing the origin of the different behavior reported in the literature will require further experimental and theoretical work.

The observed significant and mounting viscous dissipation demonstrates that the particulate nature of microgel suspensions remains dominant up to the highest packing fractions studied. Despite the close similarities of the lossy behavior with dense emulsions for $\zeta$ between 0.64 and 0.9, we find that the microgel systems enter a distinctly different regime with respect to dissipation at higher packing fractions, not accessible to emulsions, due to the effects of the interpenetrating corona chains. While storage moduli $G'(\omega)$ are nearly independent of frequency $\omega$, both for emulsions and microgels, up to the maximum possible packing density, the loss spectra $G''(\omega)$ vary quite significantly with concentration and the magnitude of $G''(\omega)$ increases even more rapidly with $\zeta$ than the storage modulus above $\zeta_c$. Indeed, superresolution microscopy suggests that faceting and weak interpenetration opens up new pathways for dissipation which explains the rising loss modulus in the overpacked regime and again highlights the essential role played by the particulate nature of the microgel suspensions and the microgel structure even at the highest packings.

## Methods

**Microgel synthesis**. pNIPAM microgels were synthesized by free radical precipitation polymerization to obtain micron sized neutral particles as described in ref. [38] and references therein. This standard protocol, used in a majority of studies on thermosensitive microgels, is known to produce inhomogeneous particles with a dense core surrounded by a fuzzy shell or corona[22]. Other, more recent synthesis approaches, use starved feed conditions to produce much more homogeneous particles without dangling ends which are not subject of this study and could be addressed in future work[60]. The synthesis was carried out in a round bottom three-necked flask, equipped with a magnetic stirrer, a reflux condenser and a gas inlet. N-isopropylacrylamide (Acros Organics, Sigma-Aldrich, 99%), NiPAM, is used as the monomeric unit. N,N-Methylenebis(acrylamide) (Sigma-Aldrich, 99%), BIS, is used as a cross-linker and N-(3-aminopropyl)methacrylamide hydrochloride (Polysciences, Sigma-Aldrich), APMA, as a co-monomer. The latter is used in order to incorporate free amine groups into the microgels to later be used as conjugation points for fluorescent dyes. NiPAM is recrystalized twice in hexane for purification, while all other chemicals are used as received. The solution is left to degas under an inert atmosphere of Nitrogen for 40 min before raising the temperature to 70 °C, a temperature at which pNIPAM is insoluble in water. The initiator, 2,2-Azobis(2-methylpropionamidine) dihydrochloride (Sigma-Aldrich, 97%, 0.0365 g), AAPH, is dissolved in 5 g of $H_2O$ prior to addition to the reaction mixture. Once this is done the solution starts to become turbid within approximately 5 min as the microgels begin to grow. At this point an injection pump containing the APMA solution is started with an addition rate of 0.5 ml/min. The synthesis is carried out for 4 h, then the reaction mixture is left to cool at room temperature overnight under constant stirring. The concentrations of the reagents are 126.9 (mmol/L) NiPAM, 6.58 (mmol/L) BIS, 1.14 (mmol/L) AAPH, and 0.37 (mmol/L) APMA. The solution is subsequently filtered to remove aggregates and, in order to remove unreacted species, 4 cycles of centrifugation, removal of supernatant and resuspension in water are performed. The resulting microgels have a degree of cross-linking of 4.9 mol% and a co-monomer concentration of 0.27 mol %, assuming complete consumption of all reagents. The swollen microgels at $T = 22$ °C are very weakly charged due to the initiator, with a measured zeta potential of $\tilde{\zeta} \simeq 16$ mV in pure water (Brookhaven PALS zeta Potential Analyzer, Smoluchowski method) in agreement with literature data suggesting values of less than 10 mV for swollen microgels of the same type[61]. For the dyed microgels (Alexa Fluor®647) we measure a similarly small value $\tilde{\zeta} \simeq 12$ mV showing that the dye labeling does not add any significant amount of charged groups to the microgel particles. Our microgels have a total radius of $R_{tot} = 470$ nm (polydispersity 6%) at $T = 22$ °C with a core radius of $R = 380$ nm, determined by static light scattering from a dilute suspension at $T = 22$ °C[37]. We find a hydrodynamic radius $R \simeq 460$ nm both in water and MEA (mercaptoethylamine, Sigma Aldrich) aqueous solution at pH = 8 used for dSTORM. Moreover we find that, within the experimental uncertainty, the hydrodynamic radius is not affected by the addition of electrolyte up to 100 mM KCl. The radius of the collapsed microgel at $T = 40$ °C is approximately $R \simeq 250$ nm (see also Supplemental Fig. 1b).

**Preparation of dense microgel suspensions**. Dense, jammed samples were obtained by centrifugation of a dilute stock suspension and redilution with pure deionized water (rheology experiments) or 50 mM MEA (mercaptoethylamine, Sigma Aldrich) aqueous solution for dSTORM. MEA is an amino thiol which acts as an

antioxidant with chemo-sensitizing and radioprotective properties[62]. We have determined the conductivity of a 50 mM MEA aqueous solution at 22 °C to approximately 4 mS/cm. All samples were prepared starting from a stock solution having a polymer concentration $c_0 = 0.9$ wt% ($\zeta = 0.072$). This initial mass density has been determined by drying the sample overnight in a vacuum oven at 80 °C and weighing. The reproducibility of drying and weighing is rather high and the remaining uncertainty with respect to the effective volume fraction $\zeta$ of approximately ±3% is mainly due to the difficulty to quantify exactly the voluminosity $k$, see also ref. [46]. To reach higher concentrations we use a temperature controlled centrifuge set to $T = 30$ °C rotating at 13,300 rpm, imposing a force of 17,000$g$. Due to the microgels being composed of mostly solvent at temperatures $T < 33$ °C, their density contrast is very low and thus centrifugation is not very efficient to reach high densities. By heating the sample above ~33 °C, the microgel size can be reduced and density increased, significantly speeding up the concentration process by centrifugation. We place the sample, contained in a clean Eppendorf tube, in the oven at 55 °C until it turns milky white, signaling the volume phase transition. This takes at most 1 min. We then immediately centrifuge it for 3 min and remove supernatant to reach the desired final concentration $c_f$ determined by weight. Due to the low-surface charge, at high temperatures and especially at higher concentrations, microgels tend to aggregate which can result in visible flocculation. This is reversible and microgels are quickly redispersed when cooled. Due to centrifugation a concentration gradient will develop in the sample. To homogenize it, the final concentrated sample is again heated, mixed via a vortex mixer then cooled. This process is repeated until the sample appears homogeneous, typically after 1–2 cycles.

**Rheology**. Oscillatory shear measurements were performed on non-labeled microgels suspended in pure water. We use a commercial rheometer (Anton Paar MCR 502), using a cone-plate geometry (cone radius 25 mm, angle 1.0°), equipped with a solvent trap to limit evaporation during the measurement. Selected examples of frequency-dependent measurements of $G'$ and $G''$ are shown in Fig. 3, covering the $\zeta$ range from marginally jammed to deeply overpacked. We also perform measurements for a fixed frequency at selected concentrations with the addition of 50 mM MEA to mimic the conditions used for the dSTORM imaging experiments, data shown in the Supplementary Fig. 6. The results are in very good agreement over the entire range of interest. We note however that measurements in MEA at the lowest concentration of 7.3 wt% ($\zeta = 0.584$), gave a torque below the sensitivity of the instrument while for pure water we find the torque sufficient to perform a measurement. This indicates a difference in elasticity between the different solvent conditions in lowest range measured. We attribute this to dissolved ions screening the charges at the tips of the dangling polymer chains, present when using ionic initiators for synthesis, weakening interactions when microgels are in close proximity. These very weakly jammed or glassy samples are however not in the focus of the present study and shall be discussed elsewhere, using more sensitive methods, such as for example dynamic light scattering or diffusing wave spectroscopy[28,56]. Finally, we note that for the weakly elastic microgel packings we observed an unphysical small drop of $G'(\omega)$ at high frequencies for data taken at $\zeta \leq 0.78$ (probably due to residual wall slip) and have thus excluded a small number of data points from the fit in Fig. 4a) to avoid bias.

**dSTORM superresolution microscopy**. The functionalized microgels are dye-labeled using the formation of stable amide bonds following the reaction between the NHS ester and the amines present in the microgel due to the addition of the APMA co-monomer acting as conjugation points. We mix an excess amount of dye with microgels in pure water and let the solution react at room temperature for 2 h on an oscillating tray, covered with aluminum foil. Removal of unreacted dye is done by 4–5 cycles of centrifugation of the sample to concentrate the heavier microgels, removal of the supernatant and resuspension in deionized water. The resulting microgels are kept in the fridge at 4 °C. As dyes we use both the fluorophore Alexa Fluor®647 and CF680R[37] (Sigma Aldrich). For dSTORM imaging we mix trace amounts of dye-labeled microgel particles in a matrix of unlabeled microgels and suspend the mixture in a 50 mM MEA (mercaptoethylamine, Sigma Aldrich) to improve the blinking for dSTORM[38,62]. For two-color dSTORM the MEA concentration was increased to 100 mM. Adding 50–100 mM MEA raises the solvent pH to about pH = 10 and we have added HCl to reduce the pH to 8. Between 60 and 80,000 frames were recorded at 60 to 100 frames per second[37]. For two-color dSTORM we apply the spectral demixing approach which is free of chromatic aberrations since both fluorophores are excited using a single laser line $\lambda = 639$ nm. The method, described in detail in ref. [37,63], relies on using two fluorophores having significantly overlapping excitation and emission spectra. Both are then simultaneously excited and the emitted light is split by a dichroic mirror (cutoff wavelength $\lambda = 690$ nm) and imaged side by side on the same camera. The split and imaging we achieve using a commercial device, OptoSplit (Cairn Research, UK), which houses the dichroic mirror alongside adjustable mirrors to produce and align the two images. We determine the color cross-talk to about 1% only[37]. From the localization of single fluorophores we extract the coordinates and reconstruct a superresolved image with the freely available ImageJ plugin ThunderSTORM[64]. The delayed addition of the co-monomer that binds to the fluorescent dye for dSTORM imaging was implemented in order enhance the signal from the boundaries leading to a depletion of signal in the center of the particles, as shown in Fig. 2, which improves contrast but is otherwise insignificant for our

analysis. The reconstructed 2D images (composed of square pixels of edge size 15 nm) originate from a plane of ~500 nm thickness adjusted to the center of the particles (see also Supplementary Fig. 7). This follows from our image reconstruction protocol, where we set a corresponding threshold for the maximum width of the Gaussian fitting the point image. The two-color images shown in Fig. 2 are taken on dye-labeled pairs of particles located directly on the cover slide to make sure that the imaging section intersects the particle at the same height. Images of individual particles recorded in the bulk show no difference compared to the particles located at the surface as shown in ref. [37], and therefore we believe the sample cell boundary does not significantly affect the deformation and interpenetration recorded in the imaging plane.

**Edge detection and contour smoothing of particles imaged by dSTORM**. Our goal now is to obtain a faithful contour of each particle and use it for measuring different geometric features such as the particle area, area overlap and deformations. To this end we must first segment the image, that is separate the particle from the background and determine the boundary. There are several methods in image analysis to accomplish this, all with their advantages and pitfalls. More sophisticated methods involve gradients or Laplacians of the image. What characterizes an edge is a sharp change in intensity, making derivatives well suited to capture their locations. We use the Laplacian of Gaussian edge detector in Matlab (MathWorks, Inc., USA), which allows for edges to be selected based on their strength, and not on image intensity values. It also produces closed contours without branching, unlike methods based solely on thresholding. The Laplacian of Gaussian edge detection consists of the following steps: first the image is smoothed by convolution with a Gaussian kernel, then the Laplacian (second derivative in 2D) is calculated and finally the zero-crossings are found which correspond to edges. Examples for the image segmentation done with this method, starting from a dSTORM image of a typical round and a more compressed and deformed particle are shown in Supplementary Fig. 8. The two particles also display different intensities (i.e., density of blinking events detected), yet we can see that the overall shapes and sizes are well captured, with a decrease in size and deformations clearly visible for the second example. Nonetheless, the presence of significant roughness of the contour remains an obstacle in characterizing shape deformations. Visual inspection of the original deformed particle image clearly suggest five corners and nearly flat facets and this are the features to isolate by image analysis. In order to smooth the particle contours while maintaining the prominent shape features we need to eliminate small scale shape fluctuations without significantly altering the overall larger scale shape and corners. To this end we use the method of Fourier descriptors[65], which has been used for a variety of applications involving image analysis. By taking the Fourier transform of the contour different spatial frequencies can be separated, then, by appropriate filtering, a smoothed contour can be retrieved and used as a starting point for further analysis. First, coordinates of the boundary pixels with respect to the particle's center of mass are extracted from the segmented image and ordered counterclockwise. We thus obtain a discrete curve $[x_j, y_j]$ with $j = 1...N$ where $N$ is the number of pixels. Then, the coordinates are transformed into complex numbers as $z_j = x_j + iy_j$. Now the Fourier descriptors $Z(k)$ are calculated using a discrete Fourier transform

$$Z(k) = \frac{1}{N}\sum_{j=0}^{N-1} z_j e^{-i2\pi jk/N} (k = 0...N - 1). \quad (1)$$

All the spatial information pertaining to the original contour is now encoded in the Fourier descriptors and, through an inverse transform, the full contour can be retrieved. The center of mass is used initially simply for centering the boundary around the particle image. An accurate determination of it is otherwise irrelevant since the descriptors inherit translation invariance from Fourier transforms. Since fine details are encoded in higher frequency components, if those are set to 0 before back transforming, noise can be eliminated and the contour smoothed. We are essentially using a low pass filter, as commonly done in signal processing. The truncated inverse transform $z'_j$, using only $N' < N$ descriptors, is calculated as $z'_j = \sum_{k=-N'/2}^{N'/2} Z(k)e^{i2\pi jk/N} (j = 0...N - 1)$. The new contour coordinates are simply obtained as $x'_j = Re(z'_j)$ and $y'_j = Im(z'_j)$. In Supplementary Fig. 9, we can see how different choices of number of descriptors influence the resulting contour. Already with $N' = 30$ we retrieve all the details and roughness of the original contour, indicating that the eliminated descriptors are redundant. At the other extreme, setting $N' = 3$, we lose instead too much detail. The contour has four soft corners instead of the five, which can be visually identified. With $N' = 6$ we obtain a very good representation of the features we can see from the image, noise is removed while all corners are still captured. In this work we have used these settings to smooth contours.

Next, we discuss the reproducibility and statistical error of the edge detection procedure. To this end we generate synthetic dSTORM images of a disc of radius 430 nm or a similarly sized hexagon (edge length 430 nm). We randomly select points in the area of a particle and repositioned them with a Gaussian error of 30 nm. Then we apply the same procedure as described above to create an image. The contour analysis is performed on this image. As shown in Supplementary Fig. 10 the uncertainty in the location of the edge drops rapidly and reaches values of less than ±5nm for the experimental conditions $N > 10,000$ points. Additional systematic errors arise from the tradeoffs already described before such as the low-density tail, Supplementary Fig. 2,

and smoothness/geometrical accuracy, Supplementary Fig. 9. The latter will lead to some smearing of sharp edges which can also be seen in the synthetic data, Supplementary Fig. 10. From this we can estimate an imaging error of about ±10 nm at the tip of edges. The accuracy of determining the contact lines and the overlap areas shown in Fig. 2 are little affected by this as long as the bare resolution is smaller than the edge length, which is the case for dSTORM (but not for standard microscopy). In summary we estimate the reproducibility of the edge detection method in the most relevant regime, extraction of the overlap area, to about ±5 nm or about 1% of the unperturbed particle radius.

## Data availability
The data sets generated during and/or analyzed during the current study are available from the corresponding author on reasonable request.

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

## Acknowledgements

This work was supported by the Swiss National Science Foundation through projects 149867 and 169074 and benefitted from support by the National Center of Competence in Research Bio-Inspired Materials. F.S. acknowledges financial support by the Adolphe Merkle Foundation through the Fribourg Center for Nanomaterials. J.L.H. acknowledges support from NSERC. We would like to thank Veronique Trappe, Sofi Nöjd and Peter Schurtenberger for discussions and for earlier contributions setting up the dSTORM imaging of microgels. We thank Joëlle Medinger and Patricia Taladriz Blanco for help with the Zeta-Potential measurements.

## Author contributions

F.S. and G.M.C. conceived the study. F.S. supervised the study. G.M.C. did all of the rheometry experiments. The dSTORM experiments were carried out by G.M.C. and P.A. C.Z. contributed to the particle size characterization, the Zeta-potential measurements and the image analysis. J.L.H. contributed to the data interpretation. All authors contributed to the data analysis and writing of the paper.

## Additional information

**Competing interests:** The authors declare no competing interests.

