## [Peer Review File · Nature Communications]

Reviewers' comments:

Reviewer #1 (Remarks to the Author):

This manuscript certainly provides new insights into the rheological behavior of microgel particle pastes at concentrations above the liquid-solid-transition, using superresolution microscopy for direct particle imaging. However, I have some doubts if the conclusions are as general as claimed by the authors. My major critical points are:

- (i) Please specify more clearly the size and crosslinker content of the particles under investigation.
- (ii) in this context: please specify the spatial resolution of the microscopic technique, which is important concerning the interpretation of structural details in the context of relevant length scales (shell thickness vs. core size, range of interdigitation, compression).
- (iii) Seiffert and coworkers have clearly demonstrated by rheological studies of microgel pastes that particle size matters (see, for example, Kunz et al., *Colloid Polym. Sci.* 2018, 296, 1341–1352): very small soft microgels seem to show no or only a very mildly pronounced liquid-solid transition, at packing fractions above 10! How does this behavior fit to the scenario developed in this manuscript? Could structural details be explored at length scales < 20 nm, relevant for such small microgel particles?

Conclusion: The idea that interdigitation and compression strongly influence the rheological behavior of microgel pastes certainly is not new. However, this manuscript presents an interesting new framework interpreting the rheological behavior of microgel pastes at very high packing fractions in much more detail than before. Still, partially due to my criticism concerning the general applicability of the new framework, I am not convinced that the manuscript is of enough relevance and common interest to really deserve publication in *Nature Comm.*

Reviewer #2 (Remarks to the Author):

This is an interesting paper which provides a correlation of rheological properties of highly concentrated microgels suspensions with the structure of the microgels. These results are very interesting for a broad community. The results are supported by the experimental data and the different regimes are very interesting. The study seems to be closely related to a recent paper in *Science Advances*, but the connection to rheological properties is new.

Thus, I support publication, however, I am missing many experimental details, which on the one hand, would not enable a different group to reproduce the experiments and on the other hand make it impossible to compare the results to other (microgel) systems. (Some of the missing information might be given in previous publications, but the references are not sufficiently clear. (In case, the identical microgel (i.e. identical batch) was used in previous papers, then this could be mentioned clearly, and some of the points I mention below seem to be missing in previous papers as well)

Anyway, the authors need to provide a complete description of the synthesis. What ingredients (e.g. what is the "comonomer"?), what concentrations, what temperature? It is well known, that details of the synthesis affect the reaction product. How did they purify the reaction product to remove the sol fraction?

Please provide more details how the very highly concentrated suspensions were prepared? Can they comment on the homogeneity of these dense suspensions? Please mention the mass concentration as well.

Furthermore, it is not clear, if the microgel could contain ionic groups from the initiator fragments or the monomer. Scotti et al have argued that the osmotic pressure of the counterions becomes relevant at high concentration. Although, my feeling is that this would not affect the main interpretation / message of the manuscript, I suggest mentioning the different approach by Scotti et al in the introduction.

Did the authors adjust the pH (and added the cysteamine) for the rheology experiments as well? If yes, please state clearly. If not, can they comment, if the different ionic strength could be

relevant?

I am not an expert in SRFM so the following questions might be naive, (but other readers might have the same questions)

Does the dye carry ionic groups at that pH? How can one perform a two-color experiment with only one dye?

My impression is that SRFM relies on many experimental conditions with respect to the dyes and the optical setup and data evaluation, so please provide such details. I know that, e.g. the Woell group does SRFM on microgels (with different types of dyes) as well, how does the accuracy of the structure determination compare to their studies? SRFM provides information on the position of the dye, is this representative of the position of the chain segments? I guess that will depend on the synthesis.

I understand that the thickness of the "slice" in the z-direction is 500nm. If the center of two neighboring microgels is not at the same "height" (position in the z-direction): does that lead to an "apparent interpenetration" in the 2D-image? (Again, I guess that will not affect the general conclusions, but the authors are rather precise on where the different regimes occur.)

Reviewer #3 (Remarks to the Author):

"Relationship between rheology and structure of interpenetrating, deforming and compressing microgels", G. Conley, J.L. Harden, F. Scheffold

This paper studies suspensions of microgels, which are particles that can potentially compress, change shape and interpenetrate, and addresses, using rheology and super-resolution microscopy, whether and how these mechanisms affect the macroscopic viscoelasticity of the suspensions. While this is interesting, it is by far not new. Moreover, the manuscript ignores existent and relevant literature on this topic and fails to properly contextualize the results, and explain the findings in connection to what has already been published by other authors and what is known for other microgel suspensions. As a result, I cannot recommend publication. In addition, I am not at all convinced the authors provide a unified framework to understand the complex macroscopic behavior of microgel suspensions. I thus recommend rejection.

Detailed comments:

* This is not the first time the rheology of pNIPAM microgel suspensions has been determined. The results in this work must be discussed in lieu of what has already been published. Since they are not the first to report these measurements, similarities and differences with prior published work must be discussed.

* In reading page 1, I find many dramatic omissions in recognizing what other authors have done. This is the case nearly every time there are citations in this page. Seminal works by Cloitre on deswelling at high concentrations and the effects on suspension rheology are not cited nor discussed. He and his group were the first to report microgel faceting and osmotic deswelling due to ions. Recent work by Sprakel on deswelling at high concentrations is not discussed. Works by Fernandez-Nieves on many aspects of microgel suspensions that are discussed in the introduction are not referred to at all. Most citations are either to work done by the authors of the paper or their collaborators, even if they were not the ones to first discuss or find or do what is stated in the text.

Near the end of the second paragraph, the authors write "there still exists no widely accepted framework that encompasses the entire range of packing densities". They must be more explicit. What is it exactly that is not known? What is the specific open question? Discussing this in lieu of existent work is of utmost importance.

A bit below, they refer to an "unsatisfactory situation". Again, they must be more specific.

* Page 2. What initiator was used in the synthesis of the microgels? There is work suggesting that

the type of initiator plays a role in the structure of the particles that result. A little below, they cite Ref. [25] (which is work by some of the authors in this paper) in connection to ζ . However, they were not the ones to first connect ζ and c using viscometry. It was Richtering.

* Related to their pNIPAM particles. The surface charge coming from the initiator and associated counterions could play a role in the results, as discussed by other groups. For instance, they report an initial shrinking of the microgels. Why does this happen instead of interpenetration? If the microgels they use have loose peripheries, why does interpenetration not happen when they come in contact? Why do they, instead, shrink? Could the ions be playing a role? This must be discussed. How do they then know these ions do not play a role even at higher ζ ? In addition, if the microgels are only compressing (or mostly doing so) in this ζ range (from ζ_{cp} to 1.2), why does G' increase so dramatically in this range? If, due to this compression, the actual volume fraction does not increase much, and there is not much interpenetration and shape deformation, why would G' increase so much? This is surprising and must be discussed. Similarly, why would G' level off after $\zeta \sim 1.2$ if interpenetration increases? Wouldn't one expect exactly the opposite? This must also be discussed.

* The oscillatory rheology presented in Fig. 2 extends down to 0.1 rad/s. I believe data at lower frequencies is needed to address whether there is a long-time relaxation or not. Data down to 0.001 rad/s should be easily accessible with the rheometer they use and hence I do not see why it is not presented. Ref. [22] presented oscillatory rheology with a different type of microgel. What is the difference with their results?

At the end of page 4, they affirm that below jamming, the measured elastic modulus is of order kT/R^3 . Can they provide numbers to support this claim? I believe the first work to observe this with Brownian emulsions was by Mason & Weitz; this work should be cited.

* The leveling off of G' with ζ has been seen before; Cloitre reported this, as well as others. These papers must be discussed and cited.

* Page 6. When discussing the initial compression regime, the authors refer to particles "predominantly interacting via their brush-like coronas". But then, why do they compress? Why don't the particles interpenetrate first?

A bit below, the authors relate the linear growth of G' with ζ to the particles becoming more homogeneous. Why? And how can this be confirmed with light scattering? Shouldn't scattering techniques with lower-wavelength-probes be used, such as neutrons? This will allow "looking" inside the particles to really probe the internal structure. There is significant work quantifying the internal structure of microgels. What do they find in these works? Do they find homogeneous microgels at high density or that the structure changes as the particles deswell? This is relevant, as deswelling also happens as density increases. These works should be taken into account.

* Page 7. What's the physics of the $\sqrt{\omega}$ term in G' ?

A bit after this, the authors refer to the increase in the liquid film confined between microgels. Cloitre has done significant work where this is discussed. However, the authors completely ignore this prior work.

A bit below, the authors again refer to the brush-like coronas to interpret the initial compressive regime. As mentioned above, I do not understand why the particles compress and do not interpenetrate. The authors must explain this.

At the end of Page 7, the authors refer to the osmotic pressure of the corona and of the core? What are these osmotic pressures? And how can they be different in equilibrium?

* Page 9, before the conclusions. The authors refer to an exponential rise. I am not sure the range of the data allows this to be established. The statement must be supported by data.

* Page 10. I really find it surprising that the seminal works of Cloitre where he refers to microgel

faceting are not cited.

* Page 11. What does the dye do to the particles? Does it change their peripheral charges? What other effects does it have? And are these effects different depending on the type of dye? How has all this been tested and quantified?

A bit below, when discussing the reconstructed 2D images. How many pairs of particles have the authors considered? What are the statistics in their measurements? This must be provided. Otherwise, it is impossible to judge the significance of the images shown in Fig. 1.

* Page 11. How was the pH adjusted to 8 with HCl? Was it even more basic than 8 to start with? Why? This must be explained. Also, what is the ionic content of the samples after pH adjustment? Is there salt?

* I have liked very much the discussion of G'' in the work. To me, this is perhaps the most significant aspect of what is presented in the paper, as it adds to what we already know about what the authors study in this work.

Summary of changes made:

- We have substantially rewritten the abstract and the made changes to introductory part, added references and tried to be more specific about our new findings. We are now putting our main emphasis on the new data on the dissipative losses and comparison to dSTORM imaging data.
- We have added the new Figure 1 to illustrate the advantages offered by dSTORM superresolution microscopy applied to microgels. We have done so also in order to improve the readability and accessibility of the manuscript for a broader audience.
- We have moved the old Fig 3 c) to the supplementary materials and have replaced this panel by a sketch of the interface between touching microgels. We did this to clarify our main idea of brush compression and lubrication followed by interpenetration. We believe this will make our work and ideas more easily accessible to the reader.
- We have substantially expanded the discussion of the microscope technique in the methods section and added more data and images to the new supplementary information (SI) file.
- We have significantly expanded the methods section about the ‘microgel synthesis’ and details about all main chemicals used are now given.
- We have slightly modified the indicative values in Figure 2 where cores touch ($\zeta \approx 1.1$) and where the packing becomes homogeneous ($\zeta \approx 1.9$) to be more in line with the discussion in the text.
- We adjusted our claims (of novelty) concerning the ‘unified framework’ for all microgels and make it clear from the beginning that we do not address small (<approx. 200nm) particles or ionic microgels.
- Page 6: We have added in-line the complete expression for $G'(\zeta)$ derived in Scheffold et al, PRL **104**, 128304 (2010)
- We have added a number of references to the work by M. Coitre, M. Cloitre and R. Seth, J. Sprakel, S. Seiffert, A. Fernandez-Nieves and others to improve our referencing of previous key contributions to the topic.
- Cysteamine and MEA are synonyms for the same chemical compound (mercaptoethylamine) used to improve dSTORM imaging (see ref [51]). In the revised manuscript we only use ‘MEA’.
- We have included a number of control measurements (existing and new ones) to verify that the influence of charges and the MEA is negligible for the range of densities studied. We have included hydrodynamic and rheology comparative measurements with and without MEA (Fig. S1b and Fig. S6a). We have measured the zeta-potential of the microgels with and without dye in pure water - which as we find is very small, around 15mV.

Next page: Reply to the Reviewers' comments- NCOMMS-18-33348 'Relationship between Rheology and Structure of Interpenetrating, Deforming and Compressing Microgels', Conley et al.

Reviewer #1 (Remarks to the Author):

This manuscript certainly provides new insights into the rheological behavior of microgel particle pastes at concentrations above the liquid-solid-transition, using superresolution microscopy for direct particle imaging.

We thank the reviewer for the positive initial assessment and we would like to address the more critical remarks in more detail below.

However, I have some doubts if the conclusions are as general as claimed by the authors. My major critical points are:

(i) Please specify more clearly the size and crosslinker content of the particles under investigation.

We have substantially expanded the method section, describing the particle synthesis and all details are now given.

(ii) in this context: please specify the spatial resolution of the microscopic technique, which is important concerning the interpretation of structural details in the context of relevant length scales (shell thickness vs. core size, range of interdigitation, compression).

We have substantially expanded the discussion of the microscope technique and discuss the resolution in the main text (for details see our 2016 article in *Colloids and Surfaces A* 499, pages 18-23), and added more information, data and images to methods section and the new supplementary information (SI) file. Briefly: The lateral optical resolution of our dSTORM implementation is approx. 30nm (Conley et al. *Colloids and Surfaces A* 499, pages 18-23). Similar to the problem of tracking the center of mass of an isolated particle this bare resolution however is not indicative of our ability to detect the compressed particle size and range of interpenetration. We use an edge detection method to determine the contour and interdigitation (see Figure 1,2, methods section and Supplementary Figure S9). To this end we analyze individual particles or pairs labelled with a different dye (two-color dSTORM). The size polydispersity we obtain for individual particles based on this method is about 5% (standard deviation to the mean) and thus comparable to the intrinsic size distribution measured with static light scattering (see SI, Fig S1). We note that the edge detection method does not capture the real interface of the swollen coronas dangling ends because of the extremely low density (see also new figure S2 in SI). However, at higher densities the corona is compressed and in this regime, where interpenetration gradually sets in (see Fig 4 c and d), the edge detection approach accurately captures the particle boundary and particle-particle interdigitation and overlap. Detecting and quantifying the amount overlap area is clearly feasible but affected by some statistical error. The data shown in Figure 4 d) is composed of the data obtained from several pairs of particles (full data set shown in SI, Fig S5b) and the error bars reflect this statistical error. Moreover, our aim is not to quantify the overlap volume exactly [a measure that is anyhow convoluted with the projection from 3D to 2D] but to show that a) there is significant overlap at high concentrations and b) interdigitation has a major impact on the dissipative losses expressed by G'' or $\tan \delta$, as shown in Figure 4 d. We note that a 3D reconstruction of particle-particle overlap is currently not within our reach.

(iii) Seiffert and coworkers have clearly demonstrated by rheological studies of microgel pastes that particle size matters (see, for example, Kunz et al., Colloid Polym. Sci. 2018, 296, 1341–1352): very small soft microgels seem to show no or only a very mildly pronounced liquid-solid transition, at packing fractions above 10! How does this behavior fit to the scenario developed in this manuscript?

We thank the reviewer for pointing out this interesting work which was published during the preparation and initial submission of our manuscript and thus escaped our attention. We have added the reference to the revised manuscript as well as the reference to Menut et al, Soft Matter 8, 156 (2012) entitled 'Does size matter? Elasticity of compressed suspensions of colloidal and granular scale microgels. We do agree with the reviewer that the reported rheological behavior of small and weakly cross-linked microgels is often found to be qualitatively different from the case we wanted to address in our work. We appreciate and acknowledge the work mentioned on smaller particles, but we cannot really add much to the discussion based on our work. **We have thus adjusted the text to highlight more clearly the scope of our study and have removed the claim for a unified rheological framework spanning all types of microgels.** It is obvious that the picture we put forward – a highly cross-linked core with radius $R \gg \xi$ decorated with a brush-like corona of thickness $L_0 \gtrsim 0.1R$ will break down as soon as $R \rightarrow \xi$. The mesh size or correlation length ξ depends on the cross linker density and can range from 5-20nm. Therefore, weakly cross-linked microgels with a hydrodynamic radius of 50-100nm are clearly out of the range and the even more strongly cross linked microgels of this size are just barely within the range of $R > \xi$.

Could structural details be explored at length scales < 20 nm, relevant for such small microgel particles?

Indeed, we do believe that dSTORM can shed light on the behavior of 'small and soft microgels'. Clearly the two-color dSTORM technique that we successfully applied to image nanoscale interdigitation of the coronas of our micron sized microgels could also reveal the interdigitation of smaller microgels. Additional sensitivity could be gained by studying spatial fluctuations in the dSTORM image even when the particle are so small that edges cannot be delineated any more. It's beyond the scope of the present work but we would be interested to pursue this important question in future work.

Conclusion: The idea that interdigitation and compression strongly influence the rheological behavior of microgel pastes certainly is not new. However, this manuscript presents an interesting new framework interpreting the rheological behavior of microgel pastes at very high packing fractions in much more detail than before. Still, partially due to my criticism concerning the general applicability of the new framework, I am not convinced that the manuscript is of enough relevance and common interest to really deserve publication in Nature Comm.

We thank Reviewer 1 for his or her comments and suggestions. We agree that the applicability of our work is mainly given for larger microgels and have changed the manuscript accordingly. Nonetheless, our results are still highly relevant for a large number of microgel systems ranging from the colloidal to the granular regime and should thus deserve publication in Nature Communications. **We would also like to emphasize that the lossy mechanism suggested in the manuscript and corroborated by our rheology**

and dSTORM data (brush sliding with no static friction followed by interdigitation and friction, Figure 3) is new and innovative and has not been reported before, to our best knowledge.

Smaller microgels have been subject to a lot of attention as well but reaching a more comprehensive understanding of their physics and rheology will require additional work that could certainly benefit from the advent of super-resolution techniques for microgel imaging put forward in our present and previously published work.

Reviewer #2 (Remarks to the Author):

This is an interesting paper which provides a correlation of rheological properties of highly concentrated microgels suspensions with the structure of the microgels. These results are very interesting for a broad community. The results are supported by the experimental data and the different regimes are very interesting. The study seems to be closely related to a recent paper in Science Advances, but the connection to rheological properties is new.

We thank the reviewer for the positive assessment and we would like to address the technical remarks and comments below.

Thus, I support publication, however, I am missing many experimental details, which on the one hand, would not enable a different group to reproduce the experiments and on the other hand make it impossible to compare the results to other (microgel) systems. (Some of the missing information might be given in previous publications, but the references are not sufficiently clear. (In case, the identical microgel (i.e. identical batch) was used in previous papers, then this could be mentioned clearly, and some of the points I mention below seem to be missing in previous papers as well)

Anyway, the authors need to provide a complete description of the synthesis. What ingredients (e.g. what is the "comonomer"?), what concentrations, what temperature? It is well known, that details of the synthesis affect the reaction product. How did they purify the reaction product to remove the sol fraction?

Please provide more details how the very highly concentrated suspensions were prepared? Can they comment on the homogeneity of these dense suspensions? Please mention the mass concentration as well.

We have added detailed information to the methods section addressing the questions raised by the referee. We thank him/her for pointing this out. The mass concentration is ζ/k , with $k=0.08$ given on page 3. Thus $\zeta=0.64$ corresponds to 8 wt/wt% - we have added this information explicitly on page 4 to illustrate how to convert effective volume fraction to mass density in our text and plots. In order not to affect the readability of the manuscript we refrained from using both c and ζ throughout the text.

Furthermore, it is not clear, if the microgel could contain ionic groups from the initiator fragments or the monomer. Scotti et al have argued that the osmotic pressure of the counterions becomes relevant at high concentration. Although, my feeling is that this would not affect the main interpretation / message of the manuscript, I suggest mentioning the different approach by Scotti et al in the introduction.

We added the Scotti, Andrea, et al. "The role of ions in the self-healing behavior of soft particle suspensions." *PNAS* 113.20 (2016): 5576-5581. We agree with the referee that in our fairly monodisperse and very weakly charged microgel system the scenario by Scotti et al may not affect the message of the manuscript. As a control we have measured the zeta-potential of the microgels with and without dye in pure water - which as we find is very small, approx. 15mV. At this point we cannot add to this discussion but would be interested to pursue this question in more detail in future work.

Did the authors adjust the pH (and added the cysteamine) for the rheology experiments as well? If yes, please state clearly. If not, can they comment, if the different ionic strength could be relevant?

Cysteamine and MEA are synonyms for the same chemical compound used to improve dSTORM imaging (see ref [51]). In the revised manuscript we exclusively used the name MEA.

The rheology experiments were done using unlabeled microgels suspended in pure water. We did not control pH. However, we have verified that the addition of MEA (for dSTORM imaging at pH8) does not change neither the swelling of microgels under dilute conditions nor the rheology of the microgel suspensions. This is now all stated in the revised Methods section and the comparison with/without MEA is included in the new SI (Figs S1b and S6a). Since the particles are swollen and only weakly charged we only see a very small influence of MEA at the lowest density studied. This is now discussed in the methods section and we state that “These very weakly jammed or glassy samples are however not in the focus of the present study and shall be discussed elsewhere”.

I am not an expert in SRFM so the following questions might be naive, (but other readers might have the same questions) Does the dye carry ionic groups at that pH?

The total dye content is controlled by the amount APMA molecules present (binding to the dye), which is less than 0.3 mol%, an order of magnitude smaller than the APMA crosslinker content. Due to the low density of dye we have no reason to suspect that charges on the dye will affect the microgel properties. We have verified in earlier work (Conley et al., *Colloids and Surfaces A* 499, pages 18-23) that the swelling behaviour of labelled and unlabelled microgels (of the type studied here) are the same. Moreover we did additional zeta potential measurements of the swollen microgels in pure water (with and without dye) and find a value of about 15mV suggesting that the particles are indeed only very weakly charged.

How can one perform a two-color experiment with only one dye?

We use two different dyes. To make this clear we have added a remark to the caption of Figure 2 and more detailed information about the two-color dSTORM in the methods section.

My impression is that SRFM relies on many experimental conditions with respect to the dyes and the optical setup and data evaluation, so please provide such details. I know that, e.g. the Woell group does SRFM on microgels (with different types of dyes) as well, how does the accuracy of the structure determination compare to their studies? SRFM provides information on the position of the dye, is this representative of the position of the chain segments? I guess that will depend on the synthesis.

We would like to thank the referee for pointing this out. We have added substantially more detailed information about the two-color dSTORM to the method section and the supplemental material. We now state in the main text that the lateral optical resolution of our dSTORM implementation is 30nm as shown in our earlier work. The match between polymer density distribution in a swollen microgel (from light scattering) and SRFM data, using the same experimental protocols, was demonstrated and discussed in detail in our first paper on SRFM, our ref. 30: Conley et al. *Colloids and Surfaces A* 499, pages 18-23). The first article of Wöll and coworkers reports similar findings, our ref 31.

I understand that the thickness of the "slice" in the z-direction is 500nm. If the center of two neighboring microgels is not at the same "height" (position in the z-direction): does that lead to an "apparent interpenetration" in the 2D-image? (Again, I guess that will not affect the general conclusions, but the authors are rather precise on where the different regimes occur.)

We thank the referee for pointing this out. We have added the following two sentences to the methods section to clarify this point: "The two-color images shown in Figure 2 are taken on dye-labelled pairs of particles located directly on the cover slide to make sure that the imaging section intersects the particle at the same height. Images of individual particles recorded in the bulk show no difference compared to the particles located at the surface as shown in [32]."

Reviewer #3 (Remarks to the Author):

“Relationship between rheology and structure of interpenetrating, deforming and compressing microgels”, G. Conley, J.L. Harden, F. Scheffold

This paper studies suspensions of microgels, which are particles that can potentially compress, change shape and interpenetrate, and addresses, using rheology and super-resolution microscopy, whether and how these mechanisms affect the macroscopic viscoelasticity of the suspensions. While this is interesting, it is by far not new. Moreover, the manuscript ignores existent and relevant literature on this topic and fails to properly contextualize the results, and explain the findings in connection to what has already been published by other authors and what is known for other microgel suspensions. As a result, I cannot recommend publication. In addition, I am not at all convinced the authors provide a unified framework to understand the complex macroscopic behavior of microgel suspensions. I thus recommend rejection.

We appreciate that the referee finds our study and the problem we address interesting. We acknowledge some shortcomings but we also regret that the referee is not able to give a more favorable assessment of our work in the first place.

We appreciate that the referee has *‘... liked very much the discussion of G'' in the work. To me, this is perhaps the most significant aspect of what is presented in the paper, as it adds to what we already know about what the authors study in this work.’*

In summary, we substantially improved the account of previous work, add more detailed information to the methods section, included additional control experiments and we now emphasize the discussion of G'' while adjusting our claims (of novelty) concerning the ‘unified framework’. We fully agree that the ‘macroscopic behavior of microgel suspensions’ is complex and that our study only provides a framework for the type of weakly charged larger microgels studied (albeit a very common one).

Detailed comments:

* This is not the first time the rheology of pNIPAM microgel suspensions has been determined. The results in this work must be discussed in lieu of what has already been published. Since they are not the first to report these measurements, similarities and differences with prior published work must be discussed.

* In reading page 1, I find many dramatic omissions in recognizing what other authors have done. This is the case nearly every time there are citations in this page. Seminal works by Cloitre on deswelling at high concentrations and the effects on suspension rheology are not cited nor discussed. He and his group were the first to report microgel faceting and osmotic deswelling due to ions. Recent work by Sprakel on deswelling at high concentrations is not discussed. Works by Fernandez-Nieves on many aspects of microgel suspensions that are discussed in the introduction are not referred to at all. Most citations are either to work done by the authors of the paper or their collaborators, even if they were not the ones to first discuss or find or do what is stated in the text.

We have significantly revised the citations and the discussion of previous work - within the constraints set by a short communication paper. In the original submission 7 out of 42 citation were to our own

work (certainly not 'most'). To gain space for additional citations (as rightfully requested by the referee) we have reduced self-citations to the 3 articles which are indispensable.

Near the end of the second paragraph, the authors write "there still exists no widely accepted framework that encompasses the entire range of packing densities". They must be more explicit. What is it exactly that is not known? What is the specific open question? Discussing this in lieu of existent work is of utmost importance.

A bit below, they refer to an "unsatisfactory situation". Again, they must be more specific.

We have substantially rewritten this introductory part, added references and tried to be more specific. We hope the revised version addresses the concerns of the referee and we thank him or her for pointing this out to us.

* Page 2. What initiator was used in the synthesis of the microgels? There is work suggesting that the type of initiator plays a role in the structure of the particles that result. A little below, they cite Ref. [25] (which is work by some of the authors in this paper) in connection to ζ . However, they were not the ones to first connect ζ and c using viscometry. It was Richtering.

We have significantly expanded the methods section about the 'microgel synthesis' and details about the synthesis and all main chemicals used are now given. We did not refer to [25] because of ζ but only because more details are given in our previous work. Figure 1 in this previous paper from 2017 shows how the swelling ratio $k=0.08$ was obtained from dynamic particle tracking (not viscosimetry). We also cite several key papers by Richtering and coworkers, our refs. 14,16, 18,19,34.

* Related to their pNIPAM particles. The surface charge coming from the initiator and associated counterions could play a role in the results, as discussed by other groups. For instance, they report an initial shrinking of the microgels. Why does this happen instead of interpenetration? If the microgels they use have loose peripheries, why does interpenetration not happen when they come in contact? Why do they, instead, shrink? Could the ions be playing a role? This must be discussed. How do they then know these ions do not play a role even at higher ζ ?

We claim that initially there is little or no interpenetration and the low density (brush like) corona of the particles is compressed. We prefer calling this process compression (due to packing) and not shrinking or de-swelling but we know that these terms are not always clearly distinguished in the literature. We would talk about shrinking or deswelling for a process that is driven e.g. by ionic effects (see e.g. Scotti et al. *PNAS* 113.20 (2016): 5576-5581).

The corona layer (dangling ends) is not (or only very weakly) crosslinked and expands from the cross-linked core into the solution (obviously the transition is not abrupt but gradual so this naturally is a simplified approach). It is well known that polymer brushes do not interpenetrate down to compression ratios of 0.1-0.15 as we have stated in the text. See also (cited by us) J. Klein, E. Kumacheva, D. Mahalu, D. Perahia, and L. J. Fetters, *Nature* 370, 634 (1994) and other work in the field. We have added a new reference showing experimentally that brushes compress and do not interpenetrate (D. J. Mulder and T. L. Kuhl, *Soft Matter* 6, 5401 (2010), unless the compression is very strong.

We have no indication that for our microgels ions released by residual charged groups play a significant role. We have presented a coherent scenario for a very common type of thermosensitive microgel (very weakly charged) with no need to discuss ion release and no evidence for ion induced shrinking or de-swelling. We added information about the measured zeta potential without dye (16mV) and with dye (12mV) under the conditions of our experiment (22C) and comparison data between microgels in pure water and in 50mM MEA aqueous suspension for the elastic modulus and the hydrodynamic radius, showing little to no differences.

In addition, if the microgels are only compressing (or mostly doing so) in this ζ range (from ζ_{cp} to 1.2), why does G' increase so dramatically in this range? If, due to this compression, the actual volume fraction does not increase much, and there is not much interpenetration and shape deformation, why would G' increase so much? This is surprising and must be discussed. Similarly, why would G' level off after $\zeta \sim 1.2$ if interpenetration increases? Wouldn't one expect exactly the opposite? This must also be discussed.

It is unfortunate that apparently we could not explain well enough our model and scenario to convince the referee. We have replaced the panel Fig 4 c (old panel now included as Fig S6b) with an illustration to make our point more clear and understandable as we hope.

We show that, first the brush-like interfacial layer is compressed and the highly cross-linked core of size R remains unchanged (for direct evidence see also new Fig S5a). Since the polymer (and fluorophore) density is very low we cannot monitor the shape deformation of the brush-like corona using dSTORM. This is clearly mentioned in the figure 2 caption and highlighted by solid lines in Figure 2 (left).

The repulsive pair-pair interactions due to brush compression are initially weak but stiffen very fast as explained by the Alexander-de Gennes brush interaction model [see our PRL 104, 128304 (2010) or S. Alexander, J. Phys. (Paris) 38, 983 (1977); P.G. de Gennes, Macromolecules 13, 1069 (1980)]. To make this more clear we have added on page 6 (in-line) the complete expression for $G'(\zeta)$ derived in Scheffold et al, PRL 104, 128304 (2010).

The brush and the cross-linked core act like a pair of coupled springs and the weaker spring dominates (the spring constant of a pair of spring in series is the reciprocal of the sum of their reciprocal). Once the brush stiffness exceeds the core stiffness the latter dominates (see also the recent work by Bergman, et al. "A new look at effective interactions between microgel particles." Nature Communications 9.1 (2018): 5039 proposing a more fine-grained approach).

For $\zeta > 1.2$ the particles are faceted and fill space more and more homogeneously (for direct evidence see new Figure S3 b). Adding more particles leads to compression which explains the linear increase in elasticity (or levelling-off in a log plot). This mechanism is not strongly affected by moderate levels of interpenetration and the concentration dependence gradually crosses over at around $\zeta = 1.9$ to what is expected for homogeneous polymer gel – the elasticity increases linearly with the mass density.

Please note that in the present manuscript we apply the knowledge about the nanoscale structure, previously obtained using dSTORM (see <http://advances.sciencemag.org/content/3/10/e1700969>), to gain better understanding of the rheology of dense microgel suspensions. In the revised manuscript we have added significantly more detailed information to the methods section and the SI discussing the experiment, the microgel synthesis/characterization as well as the dSTORM analysis.

* The oscillatory rheology presented in Fig. 2 extends down to 0.1 rad/s. I believe data at lower frequencies is needed to address whether there is a long-time relaxation or not. Data down to 0.001 rad/s should be easily accessible with the rheometer they use and hence I do not see why it is not presented. Ref. [22] presented oscillatory rheology with a different type of microgel. What is the difference with their results?

We did not record data below 0.1 rad/s. We do not have enough sample of the same batch to redo the experiments. We also do not believe it is necessary because our aim is not to look for ultra-slow terminal relaxation process and ageing. We think this is an interesting question but it's clearly not one we're addressing here.

At the end of page 4, they affirm that below jamming, the measured elastic modulus is of order kT/R^3 . Can they provide numbers to support this claim? I believe the first work to observe this with Brownian emulsions was by Mason & Weitz; this work should be cited.

The emulsion work by Mason and Weitz did not study glassy behaviour (but what is nowadays called jamming) and we believe the referee refers to this paper ?

Mason, T. G., and D. A. Weitz. "Linear viscoelasticity of colloidal hard sphere suspensions near the glass transition." *PRL* 75.14 (1995): 2770.

We rephrased the sentence mentioned by the referee slightly and we have added the citation to the work of Mason and Weitz and in turn removed the self-citation to the work by Mason, Scheffold et al (*Journal of Physics: Condensed Matter* 25, 502101 (2013)).

We note that kT/R^3 is a manifestation of caging picture of a colloidal glass. Probably the work of Mason and Weitz was not the first to show this but we agree the work is important and pioneering and have thus followed the suggestion by the referee.

* The leveling off of G' with ζ has been seen before; Cloitre reported this, as well as others. These papers must be discussed and cited.

Unfortunately the referee has overlooked that this is already the case. On page 6 of the original submission we write "Instead of the divergence of $G'(\zeta)$, predicted by the brush model we find, at higher concentrations, a slower, linear increase of elasticity as a function of packing fraction. This is in agreement with several previous studies on dense microgel packings [23, 31, 33–35] and can be attributed to the finite softness of the microgel core." Reference 31 in the original submission (now [40]) is the work by C. Pellet and M. Cloitre (*Soft Matter* 12, 3710 (2016)).

In the revised version we tried our best to cite the relevant work in the field but we cannot cite all relevant articles by an author or a group while still keeping the total number of citations reasonable. As our ref 1 we also cite the book on 'Microgel suspensions: fundamentals and applications' (John Wiley & Sons, 2011) which gives a good overview over the literature prior to 2010.

* Page 6. When discussing the initial compression regime, the authors refer to particles "predominantly interacting via their brush-like coronas". But then, why do they compress? Why don't the particles interpenetrate first?

As discussed above dense polymer brushes on flat surfaces do not interpenetrate but compress down to compression ratios of 0.1-0.15. This is established polymer physics knowledge and there is a huge literature starting with the work of Alexander and de Gennes in the 1970's. Unfortunately we cannot cite all the relevant work but we added another more recent reference [48]. We have cited the work by Klein and Kumacheva [47] because it showed both i) that brushes do not interpenetrate unless they are strongly compressed and b) as a consequence the interfaces are lubricated (no static friction). Based on this picture we argue that interpenetration starts when the brush is highly compressed and the density of the compressed brush is similar to the density of the core because at this point the energy penalty for interpenetration vanishes. See also our new illustration in Fig 4c and Fig 6 in our new ref 48: D. J. Mulder and T. L. Kuhl, *Soft Matter* 6, 5401 (2010).

A bit below, the authors relate the linear growth of G' with ζ to the particles becoming more homogeneous. Why? And how can this be confirmed with light scattering? Shouldn't scattering techniques with lower-wavelength-probes be used, such as neutrons? This will allow "looking" inside the particles to really probe the internal structure. There is significant work quantifying the internal structure of microgels. What do they find in these works? Do they find homogeneous microgels at high density or that the structure changes as the particles deswell? This is relevant, as deswelling also happens as density increases. These works should be taken into account.

By homogeneous we mean that the corona and the core have similar density once the corona has been compressed on the core and the core shell structure is lost. To make this clear we have revised the conclusion section and written as follows:

'At higher packing fractions, we visually observe deformation, interpenetration and compression and here the elasticity increases linearly with concentration starting at ζ_c , in agreement with the jamming picture of dense assemblies of homogeneous soft spheres [8, 10]. In this regime microgel particles have lost their core-shell structure as the soft brush-like corona has already been compressed onto the core. Eventually the faceted and interpenetrating polymeric particles fill space homogeneously.'

We are aware of the neutron scattering work e.g. by A. Fernández-Barbero, A. Fernández-Nieves, I. Grillo, and E. López-Cabarcos, *Phys. Rev. E* 66, 051803 but we do not think this kind of information probed by neutron scattering, such as the local correlation length, can be used in our study (it certainly influences the compressibility of the microgel on an absolute scale but we do not address this question). With respect to 'deswelling'. We talk about compression of soft particles due to crowding (which some people also call deswelling - see Aguiar et al *Scientific Reports* 7, 10223 (2017) - we do not). We have no indications that our particles deswell due to ionic effects.

* Page 7. What's the physics of the $\sqrt{\omega}$ term in G'' ?

We assume the referee refers to the $\sqrt{\omega}$ term in G'' (loss modulus). This expression is directly taken from the theory of Andrea Liu et al initially derived for suspensions of spheres in the absence of static friction - our reference [46] (A. J. Liu, S. Ramaswamy, T. Mason, H. Gang, and D. A. Weitz, *Physical Review Letters* 76, 3017 (1996)). As written in our text and summarized from this previous work "The random orientation of slip planes leads to a broad range of stress relaxation rates that result... "[the $\sqrt{\omega}$ term in G'']. Details of the theory and comparison to experimental data taken on compressed emulsions are given in the work by Liu et al. The model proposes that the mechanism for

dissipation is due to shearing the liquid compressed between the faceted interfaces. The assumption is that there is no static friction (for the emulsions and, **as we show here for the first time**, for the compressed microgels prior to the onset of interpenetration).

A bit after this, the authors refer to the increase in the liquid film confined between microgels. Cloitre has done significant work where this is discussed. However, the authors completely ignore this prior work.

We fully agree and thank the referee for pointing this out. We have added a sentence and citation on page 9: "Earlier pioneering work by Cloitre and co-workers already suggested that densely packed microgels develop flat facets at contact and that the thin water film trapped in between can lubricate the contacts, but a connection to the model of Liu et al. was not made [11] (Cloitre et al in PRL 90, 068303 (2003))

A bit below, the authors again refer to the brush-like coronas to interpret the initial compressive regime. As mentioned above, I do not understand why the particles compress and do not interpenetrate. The authors must explain this.

We refer to the detailed discussion about polymer brushes above. This is also in full agreement with the statement taken from the work by Cloitre, Leibler and coworkers. We also added the graphical illustration in Figure 4c) to make this point more clear.

At the end of Page 7, the authors refer to the osmotic pressure of the corona and of the core? What are these osmotic pressures? And how can they be different in equilibrium?

We fully agree and apologize for this mistake. We have rewritten this sentence: "Chains in the corona are not stretched anymore and the density of the corona and the core gradually approach each other". If chains are not stretched and the density is the same there is no energetic or entropic penalty any more for interpenetration of the outer layer of dangling polymer chains.

* Page 9, before the conclusions. The authors refer to an exponential rise. I am not sure the range of the data allows this to be established. The statement must be supported by data.

We agree and have rewritten this 'both seem to rise together toward a plateau value at large ζ '. In the caption to Figure 4d we now write 'solid line...is a guide to the eye'.

* Page 10. I really find it surprising that the seminal works of Cloitre where he refers to microgel faceting are not cited.

Indeed, we agree. We have added references to some of the key work by Cloitre and coworkers (not all, due to space) on microgel elasticity and faceting to the main text but think we do not need to cite the work again in the conclusion section.

* Page 11. What does the dye do to the particles? Does it change their peripheric charges? What other effects does it have? And are these effects different depending on the type of dye? How has all this been tested and quantified?

We do not expect the dye labeling and the associated charge to have significant influence on the particle properties. The molar concentration of co-monomer holding the dye molecule is less than 0.3%. We have tested whether adding the dye influences the swelling and structure for isolated microgel particles in our first dSTORM paper (G. M. Conley, S. Nojd, M. Braibanti, P. Schurtenberger, and F. Scheffold, *Colloids and Surfaces A* 499, 18 (2016)) and found quantitative agreement between the dSTORM and light scattering.

It is not straightforward for us to test this directly for dense packing since the labelled particles are seeded into the bulk sample at a very low density and would thus not affect the rheological properties. We have also verified that the MEA added to allow dSTORM does not affect the microgel swelling and rheological properties (see new SI, Fig S1b and Fig S6a). Moreover we have measure the zeta-potential with and without dye and find similarly small values around 15mV.

Overall the results seem to be very robust to different solvent conditions for the weakly charged microgels we study.

We also note that dyed microgels are used by others, but we fully agree with the referee that his has to be tested and verified case by case.

A bit below, when discussing the reconstructed 2D images. How many pairs of particles have the authors considered? What is the statistics in their measurements? This must be provided. Otherwise, it is impossible to judge the significance of the images shown in Fig. 1.

A more extensive dSTORM data set is presented in *Science Advances* 3, e1700969 (2017). In total we studied about 100 pairs (see SI figure 5b). In the present manuscript we have binned the results for a given density and plotted in Figure 4 d for clarity. Each point with error bar corresponds to typically 4 experiments. These are very difficult experiments. We do not claim more than what we have. There is a clear trend and what we say is that 'both [i.e. ΔF and $\tan \delta$] seem to rise together toward a plateau value'. The full data set is now also shown in the SI in Fig. S5b

* Page 11. How was the pH adjusted to 8 with HCl? Was it even more basic than 8 to start with? Why? This must be explained. Also, what is the ionic content of the samples after pH adjustment? Is there salt?

We have added the sample preparation details to the method section. MEA is basic and we have used HCl to set pH=8 for imaging. We also added this information to the methods section under 'Methods/dSTORM superresolution microscopy'.

We did not control the ionic strength during the rheology experiments, However, we checked carefully whether the addition of the MEA makes a difference and we see no significant change except for a tiny difference in modulus at the lowest density which we now discuss in the methods section. We thus conclude that charges are not relevant in our case because the particles are very weakly charged or neutral and we focus our attention on jammed packings with $G' \gg 10\text{Pa}$ (and not the weak moduli $kT/R^3 < 1\text{ Pa}$ expected around the glass transition which might indeed be influenced by the residual charges - see discussion we added to the methods section).

* I have liked very much the discussion of G'' in the work. To me, this is perhaps the most significant aspect of what is presented in the paper, as it adds to what we already know about what the authors study in this work.

We thank the referee for this remark and we agree. In the revised manuscript we have emphasized the discussion of G'' and the novelty in our work. We have rephrased or dropped claims about providing a unified picture for the elasticity of (all) microgels.

Reviewers' comments:

Reviewer #1 (Remarks to the Author):

All critical remarks, raised in my first review report, have been addressed by the authors in a satisfactory manner. Therefore, I now recommend publication of the paper in its actual version.

Reviewer #2 (Remarks to the Author):

The authors have improved the manuscript and many things are much clearer now. I have only a few small comments.

It is still not clear to me how the data are related to the previous papers by the authors in CSA (2016) and Science Advances (2017): They should clearly mention if they use the identical batch of microgels or not.

In case they used the identical batch: As images often look very similar: do (some) of the images show identical samples as in the Science Advances (2017) paper? (the three microgels on the right part in Figure 1 a appear very similar to the image in Fig 1 of Science Advances (2017) paper). Please mention in case they show identical objects.

I suggest adding information on the size of the microgels in the collapsed state to the part on microgel synthesis.

Concerning imaging microgels at the interface (Main text and Figure S7):

Obviously, the authors assume that the microgels have no attractive with the wall and thus do not adsorb and deform. I suggest that they either show experimental evidence for that or mention (in the main text) clearly that they assume no deformation.

Reviewer #3 (Remarks to the Author):

"Relationship between rheology and structure of interpenetrating, deforming and compressing microgels", G. Conley, J.L. Harden, F. Scheffold

I think the paper has greatly improved. However, I still have some comments/questions, mostly related to the answers they have provided to the referees. I also think there is some important information that should be provided in the manuscript itself, even if more details are provided in the supplementary materials.

Response to ref 1. (a) The authors quote 30 nm as the lateral resolution of their dSTORM implementation (1st page ref. 1). However, they say this does not limit their ability to detect compressed particle size and range of interpenetration. What is the resolution then? This must be explicitly stated in the paper. Furthermore, later in their reply (2nd page ref. 1) they answer positively to the question of the referee on whether they can resolve structural details below 20nm. This seems contradictory with having a lateral resolution of 30 nm, unless their detection algorithm improves this significantly. It is thus important that the authors clarify this and provide the actual resolution. (b) The fact that soft microgel suspensions do not solidify until very, very high packing fractions has indeed been seen before, even with micron sized microgels. Why is this not seen with their microgels? Addressing this is important, as it allows seeing if there is anything special with this system.

Response to ref. 2. (a) 1st page ref. 2, last question. Why would being monodisperse be a cause for the deswelling referred to the referee not being important? In addition, most of the experiments done by the authors to test whether charge or ion content is significant is done in dilute conditions, where the ionic effects reported by Scotti et.al are not important (this same arguments and issues also show up in their reply to ref. 3). What is the actual ionic concentration

in the samples used by the authors? This must be clearly stated in the manuscript itself. In addition, it should be stated whether the charged groups have a base or acid character (there are several groups, coming from the initiator and also from co-monomers) and explicitly say what the pH is and whether these groups would be charged or not in the experimental conditions. The complexity of the particles has to be clearly provided in the manuscript. And the conditions of the experiments must also be given (pH and ionic concentration). Not providing these details prevents assessing whether ionic effects at high densities could matter or not. Providing all this, will not only help address this per se, but it could also help justify that the rheology of dyed and undyed microgels is similar.

Response to ref. 3. (a) Page 2, ref. 3, last question. The authors do not answer the question. Why does shrinking happen before interpenetration? They point of the question is why. I agree that this is seen in polymer brushes, but providing an explanation for why it happens would be of value. (b) Page 4, ref. 3, second question. Can they provide the value of kT/R^3 and compare it to the G' they measure? This would support their claim on the glassy (before jamming) origin of the elasticity.

The paper has improved significantly. Once the details above have been incorporated and the questions answered, I would be happy to recommend publication.

Next page: Reply to the Reviewers' comments- NCOMMS-18-33348 'Relationship between Rheology and Structure of Interpenetrating, Deforming and Compressing Microgels', Conley et al.

Reviewers' comments:

Reviewer #1 (Remarks to the Author):

All critical remarks, raised in my first review report, have been addressed by the authors in a satisfactory manner. Therefore, I now recommend publication of the paper in its actual version.

We would like to thank the reviewer for the positive assessment and the earlier constructive comments that have helped us to improve the manuscript.

Reviewer #2 (Remarks to the Author):

The authors have improved the manuscript and many things are much clearer now. I have only a few small comments.

We would also like to thank the reviewer for the previous comments and we appreciate that the reviewer now approves our work.

It is still not clear to me how the data are related to the previous papers by the authors in CSA (2016) and Science Advances (2017): They should clearly mention if they use the identical batch of microgels or not.

We have added a sentence on page 3 to make this point clear 'The same batch of microgels was used in our earlier work, ref. [32]. '.

In case they used the identical batch: As images often look very similar: do (some) of the images show identical samples as in the Science Advances (2017) paper? (the three microgels on the right part in Figure 1 a appear very similar to the image in Fig 1 of Science Advances (2017) paper). Please mention in case they show identical objects.

We would like to thank the referee for pointing this out. The image included in the last version, although reformatted, shows some of the the same objects as in Fig 1 of our Science Advances (2017) paper (isolated microgels on a glass surface). Although this would be OK we believe it is indeed better to show a representative set of particles located in the bulk. We have thus replaced Figure 1 a) and b) with images taken in the bulk.

I suggest adding information on the size of the microgels in the collapsed state to the part on microgel synthesis.

We have added a sentence to the microgel synthesis section: 'The radius of the collapsed microgel at T = 40C is approximately $R = 250\text{nm}$, see supplemental Fig. S1b.'

Concerning imaging microgels at the interface (Main text and Figure S7):

Obviously, the authors assume that the microgels have no attractive [interaction] with the wall and thus do not adsorb and deform. I suggest that they either show experimental evidence for that or mention (in the main text) clearly that they assume no deformation.

We have added a half-sentence (underlined) to the dSTORM superresolution microscopy Methods section: Images of individual particles recorded in the bulk show no difference compared to the particles located at the surface and therefore we believe the sample cell boundary does not significantly affect the deformation and interpenetration recorded in the imaging plane.

Reviewer #3 (Remarks to the Author):

“Relationship between rheology and structure of interpenetrating, deforming and compressing microgels”, G. Conley, J.L. Harden, F. Scheffold

I think the paper has greatly improved. However, I still have some comments/questions, mostly related to the answers they have provided to the referees. I also think there is some important information that should be provided in the manuscript itself, even if more details are provided in the supplementary materials.

We would like to thank the reviewer for the much more positive assessment and also the earlier comments that certainly have helped us to improve the manuscript.

Response to ref 1. (a) The authors quote 30 nm as the lateral resolution of their dSTORM implementation (1st page ref. 1). However, they say this does not limit their ability to detect compressed particle size and range of interpenetration. What is the resolution then? This must be explicitly stated in the paper. Furthermore, later in their reply (2nd page ref. 1) they answer positively to the question of the referee on whether they can resolve structural details below 20nm. This seems contradictory with having a lateral resolution of 30 nm, unless their detection algorithm improves this significantly. It is thus important that the authors clarify this and provide the actual resolution.

We have added a sentence to the main text and moreover we added a detailed discussion and analysis using synthetic data to the methods section at the end of subsection ‘Edge detection and contour smoothing of particles imaged by dSTORM’.

Briefly summarized: a finite resolution of e.g. 30nm leads to the blurring of objects. Instead of a perfect disc (in 2D) of size e.g. $R=430\text{nm}$ one sees a disk with blurred edges. If the blurring is Gaussian (which is a good approximation), the edge will be something like an error-function. If we fit this error function using our edge detection method we can still detect the radius of the disc with perfect statistical accuracy as long as there is a sufficient number of points (sampling of the dSTORM image). For more complex objects - such as deformed, faceted microgels, it is also important to have the resolution be smaller than the particle size and the length of the long edges of the facets because otherwise there would be too much signal cross-talk around the edges or across the particle. A full analysis of these effects (for the 500nm image sections of our non-homogeneous particles projected from 3D to 2D) is quite complicated and beyond the scope of our work but we have added two (synthetic data) examples, representative of our case, to the supplemental information, Fig. S10. In conclusion, based on the analysis of representative synthetic data, we estimate the statistical error of the edge localization to about $\pm 5\text{nm}$ and we have added a mention of this value to the text.

(b) The fact that soft microgel suspensions do not solidify until very, very high packing fractions has indeed been seen before, even with micron sized microgels. Why is this not seen with their microgels? Addressing this is important, as it allows seeing if there is anything special with this System.

Unfortunately we cannot comment on this. It's not seen for our microgels, neither in many other studies using similar synthesis protocols. We believe this should be answered by the authors of the work that observe these phenomena. We believe their findings are more unexpected than ours because they

certainly require an additional deswelling mechanism to explain their finding which is not present in our system.

Response to ref. 2. (a) 1st page ref. 2, last question. Why would being monodisperse be a cause for the deswelling referred to the referee not being important?

Upon request by the reviewer 2 we have added a citation to *PNAS* 113.20 (2016): 5576-5581 in the 1st revision of the manuscript. This work by Scotti et al. deals with mixtures of large (diameter 360nm) microgels seeded in a matrix of smaller (250nm) microgels. We simply wanted to make it clear that our system is not bidisperse. We make no statements about the causes of deswelling because it is not relevant for us and should be discussed elsewhere. We absolutely see no measurable evidence for ionic effects in our system (see also below).

In addition, most of the experiments done by the authors to test whether charge or ion content is significant is done in dilute conditions, where the ionic effects reported by Scotti et.al are not important (this same arguments and issues also show up in their reply to ref. 3). What is the actual ionic concentration in the samples used by the authors? This must be clearly stated in the manuscript itself. In addition, it should be stated whether the charged groups have a base or acid character (there are several groups, coming from the initiator and also from co-monomers) and explicitly say what the pH is and whether these groups would be charged or not in the experimental conditions. The complexity of the particles has to be clearly provided in the manuscript. And the conditions of the experiments must also be given (pH and ionic concentration). Not providing these details prevents assessing whether ionic effects at high densities could matter or not. Providing all this, will not only help address this per se, but it could also help justify that the rheology of dyed and undyed microgels is similar.

We have stated that we use 'pure water' in the sample preparation methods section. We think it's beyond the scope of our work to discuss all the charged groups that might be active since we have already shown that pH or ionic effects do not play a role in our nearly neutral and swollen microgels. To (as we hope) settle this question we have measured the hydrodynamic radius of the particles using DLS at 22C for different ionic strengths up to 100mM KCl and find absolutely no influence of the electrolyte concentration on the hydrodynamic radius. We have added the plot as an inset in Fig. S1b. We note also that the elastic shear modulus does not depend on the addition of the 50mM mercaptoethylamine, Fig S6a, as mentioned earlier. In summary, we believe we have made every possible effort to show that ionic effects are not of essence here and we have described our experiments in a detailed and transparent way. We hope that the reviewer is now satisfied with our explanations.

We have added a sentence to the methods section stating that "the hydrodynamic radius is found to be unaffected by the addition of electrolyte up to 100mM KCl".

Response to ref. 3. (a) Page 2, ref. 3, last question. The authors do not answer the question. Why does shrinking happen before interpenetration? They point of the question is why. I agree that this is seen in polymer brushes, but providing an explanation for why it happens would be of value.

In the last revision we have added the article ref. [48] by D. J. Mulder and T. L. Kuhl (*Soft Matter* 6, 5401 (2010)) dedicated to this topic which gives a more up-to-date description of the phenomenon. In the

new revision we have added the review by Milner (new ref. [48], Milner, S. T. (1991). Polymer brushes. Science, 251(4996), 905-914.) where he summarizes on page 913 the results for ideal polymer brushes : “Brushes brought into contact do not interpenetrate because a chain configuration that meanders into the opposite brush is more stretched and encounters more chain units than necessary”.

A repetition of the polymer physics explanation developed in the 1970's and 1980's, parts of which are nowadays also found in textbooks, is beyond the scope of our work but the Milner review gives a very good introduction to the topic. A discussion for compressed brushes and experimental evidence to what point interpenetration occurs in strongly compressed brushes is given in the work by D. J. Mulder and T. L. Kuhl.

(b) Page 4, ref. 3, second question. Can they provide the value of kT/R^3 and compare it to the G' they measure? This would support their claim on the glassy (before jamming) origin of the elasticity.

We have added this information to the text on page 6: In our case $kT/R^3 \approx 0.005\text{Pa}$ which then crosses over to a regime governed by the jamming elasticity (see also ref. [27] and supplementary Fig. S6). We note that we cannot measure the very small moduli at the onset of the glass transition but it is expected that the elasticity rises in the glass (diverges for hard spheres at random close packing), which is discussed in ref [27] (and the new ref. [40]) and our experiments suggest that we pick up the final stages of the cross over from glassy elasticity to the jamming of soft microgel spheres.

The paper has improved significantly. Once the details above have been incorporated and the questions answered, I would be happy to recommend publication.

Would like to thank the reviewer for the comments and suggestions and we hope that with the additional changes made the manuscript is now ready for acceptance and publication.

REVIEWERS' COMMENTS:

Reviewer #2 (Remarks to the Author):

the authors have addressed my comments sufficiently.

Concerning the influence of ionic groups:

The relevance of the presence of few ionic groups is an open question and I agree with the reviewer that this point must be clearly stated in a paper. However, I agree with the authors that as the DLS data at different salt content do not reveal an influence of salt on the diameter, one can assume that charge interactions are not dominant. The dSTORM experiments are done at rather high salt and as they seem to fit to the other experiment, I agree with the authors that charge interactions are not dominant. On the other hand, the fact that "pure" water was used for rheology instead of the 50mM MEA as for dSTORM is not the best practise. Having at least a few rheology experiments done in 50mM MEA would have been of advantage.

Concerning the solidification at high microgel concentration:

Well, I guess one has to live with discrepancies between different studies. Having more studies in the literature will finally clarify what the origins for different behaviours are. I agree with the authors that they cannot explain that.

However, the authors could do two things to help the readers:

- (i) provide the experimental error for the packing fraction and add that as horizontal error bars to the figures 3b and 4 b and 4 d.
- (ii) When comparing to the results to the literature, try being specific about the range of packing fractions and the size of the microgels. Different studies might have looked at different regimes.

To conclude: I suggest that the authors do add the error bars in the packing fraction and revise the text a bit further; however I think that requesting a (the?) final explanation about the influence of ionic groups is too much.

Reviewer #3 (Remarks to the Author):

"Relationship between rheology and structure of interpenetrating, deforming and compressing microgels", G. Conley, J.L. Harden, F. Scheffold

I do not think the authors have properly addressed some of my comments. I find their reply rather dismissive. (i) After comment (b), reviewer 3 - The authors simply say they cannot comment on this. However, their first reply did comment on this. They concluded that their microgels are too large to fall within what referee 1 was asking about. In my reply, I told them that there are large microgels, of similar size to theirs, which also remain liquid-like or solidify at very high packing fractions, implying their response was not accurate. I thus do find their reply (that the question is out of the scope of the work) convincing. (ii) They do not answer the following question: "What is the actual ionic concentration in the samples used by the authors?" All they refer to are measurements in dilute conditions. This reflects they have not considered carefully the work by Scotti highlighting that ions can be of enormous importance at high concentration, even for "neutral" microgels that are only charged at their periphery. The fact that in dilute conditions, the size does not change with salt, does not preclude ionic effects to have an enormous influence at high packing fractions. Even if only a few ions are present, at high packing fractions, due to small free volume outside the particles, these ions could exert an osmotic pressure that could have large effects. Knowing the salt concentration in their samples is thus very important, as these effects could be washed out depending on salt concentration. However, the authors do not answer the question and simply dismiss it. They also mention that providing information about the sources of charge is not relevant for this work. This reflects they have not considered the comments of the referees seriously. A small discussion about this is needed to have a sense of the complexity of the particles and of what groups can contribute charges to their particles and where will these be located. In conjunction with these charges, there will be counterions in solution. In addition, knowing the background ions (from salt or buffers or anything else) is also of utmost importance.

Again, none of their studies in dilute conditions preclude the ions from being important at high packing fractions. The authors must provide this information or try to at least estimate the best they can the amount and type of ions present in their experiments. (iii) Briefly discussing the physics of compressing rather than interpenetrating increases the readability of the paper. This is an important aspect to this work. Hence, a physical explanation would help make the paper more accessible. Their answer here is again quite dismissive. That they do not find this warranted does not mean that other equally qualified readers would find this of importance.

I believe all this should be considered and taken into account. Once it is, I'll be happy to recommend publication.

Summary of changes made:

- We have added the x-error bars to Figure 3 and 4 and included more information about the size and type of microgel when comparing our results to the literature.
- We have added the following two sentences to the discussion part: “ Interestingly, in contrast to other recent studies, we find no evidence for spontaneous deswelling due to an osmotic pressure difference between the inside and the outside of the microgels as suggested by Scotti et al. Revealing the origin of the different behaviour reported in the literature will require further experimental and theoretical work.. “
- We have added about a dozen additional references. The new references were mainly added to the introduction, some in response to the comments by the referees and two additional references to our own work.

Reply to the Referee’s’ comments - ‘Relationship between Rheology and Structure of Interpenetrating, Deforming and Compressing Microgels’, Conley et al.

Reviewer #2 (Remarks to the Author):

the authors have addressed my comments sufficiently.

Concerning the influence of ionic groups:

The relevance of the presence of few ionic groups is an open questions and I agree with the reviewer that this point must be clearly stated in a paper. However, I agree with the authors that as the DLS data at different salt content do not reveal an influence of salt on the diameter, one can assume that charge interactions are not dominant. The dSTORM experiments are done at rather high salt and as they seem to fit to the other experiment, I agree with the authors that charge interactions are not dominant. On the other hand, the fact that “pure” water was used for rheology instead of the 50mM MEA as for dSTORM is not the best practise. having at least a few rheology experiments done in 50mM MEA would have been of advantage.

Rheology experiments done in 50mM were already include in the Figure S6a) and show little to no difference. A more detailed discussion of Fig S6a can be found in our first rebuttal letter:

“The rheology experiments were done using unlabeled microgels suspended in pure water. We did not control pH. However, we have verified that the addition of MEA (for dSTORM imaging at pH8) does not change neither the swelling of microgels under dilute conditions nor the rheology of the microgel suspensions. This is now all stated in the revised Methods section and the comparison with/without MEA is included in the new SI (Figs S1b and S6a). Since the particles are swollen and only weakly charged we only see a very small influence of MEA at the lowest density studied. This is now discussed in the methods section and we state that “These very weakly jammed or glassy samples are however not in the focus of the present study and shall be discussed elsewhere”.

Concerning the solidification at high microgel concentration:

Well, I guess one has to live with discrepancies between different studies. Having more studies in the literature will finally clarify what the origins for different behaviours are. I agree with the authors that they cannot explain that.

However, the authors could do two things to help the readers:

(i) provide the experimental error for the packing fraction and add that as horizontal error bars to the figures 3b and 4 b and 4 d.

(ii) When comparing to the results to the literature, try being specific about the range of packing fractions and the size of the microgels. Different studies might have looked at different regimes.

To conclude: I suggest that the authors do add the error bars in the packing fraction and revise the text a bit further; however I think that requesting a (the?) final explanation about the influence of ionic groups is too much.

We have added the error bars and included more information about the size and type of microgel when comparing our results to the literature.

Reviewer #3 (Remarks to the Author):

“Relationship between rheology and structure of interpenetrating, deforming and compressing microgels”, G. Conley, J.L. Harden, F. Scheffold

I do not think the authors have properly addressed some of my comments. I find their reply rather dismissive. (i) After comment (b), reviewer 3 - The authors simply say they cannot comment on this. However, their first reply did comment on this. They concluded that their microgels are too large to fall within what referee 1 was asking about. In my reply, I told them that there are large microgels, of similar size to theirs, which also remain liquid-like or solidify at very high packing fractions, implying their response was not accurate. I thus do find their reply (that the question is out of the scope of the work) convincing. (ii) They do not answer the following question: “What is the actual ionic concentration in the samples used by the authors?” All they refer to are measurements in dilute conditions. This reflects they have not considered carefully the work by Scotti highlighting that ions can be of enormous importance at high concentration, even for “neutral” microgels that are only charged at their periphery. The fact that in dilute conditions, the size does not change with salt, does not preclude ionic effects to have an enormous influence at high packing fractions. Even if only a few ions are present, at high packing fractions, due to small free volume outside the particles, these ions could exert an osmotic pressure that could have large effects. Knowing the salt concentration in their samples is thus very important, as these effects could be washed out depending on salt concentration. However, the authors do not answer the question and simply dismiss it. They also mention that providing information about the sources of charge is not relevant for this work. This reflects they have not considered the comments of the referees seriously. A small discussion about this is needed to have a sense of the complexity of the particles and of what groups can contribute charges to their particles and where will these be located. In conjunction with these charges, there will be counterions in solution. In addition, knowing the background ions (from salt or buffers or anything else) is also of utmost importance. Again, none of their studies in dilute

conditions preclude the ions from being important at high packing fractions. The authors must provide this information or try to at least estimate the best they can the amount and type of ions present in their experiments. (iii) Briefly discussing the physics of compressing rather than interpenetrating increases the readability of the paper. This is an important aspect to this work. Hence, a physical explanation would help make the paper more accessible. Their answer here is again quite dismissive. That they do not find this warranted does not mean that other equally qualified readers would find this of importance.

We believe we have already responded to these questions by the referee #3 in detail in our previous exchange. Since we do not see evidence for the ionic deswelling effects of the kind proposed by Scotti et al. we think it is not appropriate for us to draw conclusions on this topic in the present manuscript. Nonetheless we do agree with the referees that the influence of ionic groups can be important in some cases. Which cases will probably depend on the microgel architecture, cross link density and elasticity, residual charged groups, ionic strength etc. this is an important open questions and we - as well as others - will certainly address it in future work.

To emphasize this point we have added two sentences to the discussion part: “ Interestingly, in contrast to other recent studies, we find no evidence for spontaneous deswelling due to an ionic osmotic pressure difference between the inside and the outside of the microgels as suggested by Scotti et al. Revealing the origin of the different behaviour reported in the literature will require further experimental and theoretical work. “

I believe all this should be considered and taken into account. Once it is, I'll be happy to recommend publication.